# Beyond Extrapolation: Knowledge Utilization with Bidirectionally Inspired Auxiliary Stream for Time Series Forecasting

## Abstract

Time-series forecasting is critical in application areas such as energy, transportation, and public health. Most existing forecasters, however, are designed primarily around unidirectional inference from **history** to **target**. While this formulation has achieved strong performance in many practical scenarios, it focuses solely on the history–target link and leaves unused the structured information in how trajectories continue after the target, even though such post-target behaviour can provide a valuable inductive bias for forecasting. In a typical time series, each training example naturally forms a chain of three segments: "**history** (model input), **target** (ground-truth output), **post-target continuation**". In this work, we explicitly use the third segment as a source of auxiliary features and propose KUP-BI (Knowledge Utilization Paradigm with a Bidirectionally Inspired Auxiliary Stream), a simple non-parametric mechanism that distils continuation-style information from a train-only historical library and injects it into standard forecasting backbones. For each training chain, we extract an equal-length history window and post-target continuation window, apply a simple ratio-style operator that encodes how the continuation changes relative to its history, and store the resulting transformation together with its history in the library. Given a current input window, we extract similar historical segments from this library, aggregate their associated transformations, and apply the aggregated transformation to the current input to obtain a deterministic continuation-style auxiliary feature that summarises how similar histories tended to evolve in the training data. The input and auxiliary streams are encoded separately and fused through a lightweight feature-level gating module. This design does not introduce information beyond what is already contained in the training trajectories, but provides a structured inductive bias that helps backbones exploit typical continuation patterns rather than relying solely on parametric extrapolation. Across six benchmarks and several state-of-the-art models, KUP-BI consistently improves forecasting performance with small additional overhead.

## 1 Introduction

Time-series forecasting plays an important role in finance (Huang et al. (2024)), traffic (Zeng et al. (2023b)), weather (Lam et al. (2023)), and energy (Wang et al. (2019)). As the need for accurate predictions has grown, methods have evolved from single-step to multi-step forecasting horizons (Zhou et al. (2021); Wu et al. (2021)) and from linear to nonlinear models (Box et al. (2008); Li et al. (2023); Shao et al. (2025)). Recent work shows that deep learning models can capture complex nonlinear patterns and improve long-horizon forecasting on real-world data (Kudrat et al. (2025); Liu et al. (2025)).

Although recent deep learning forecasters have made notable progress, they typically operate under a one-way inference paradigm that focuses on learning the mapping from **history** to **target** in the natural chain "**history** (model input), **target** (ground-truth output), **post-target continuation**" (Shao et al., 2025; Wen et al., 2023). This paradigm has been highly successful across many benchmarks, especially when dynamics are relatively stable and the history is well aligned with the upcoming horizon. In more complex settings with local changes or fine-grained variability, the backbone can

still fit the training pairs well: history segments that look similar within the training set are mapped to consistent target segments. At test time, however, inputs that look similar to these training histories can still have slightly different recent dynamics, so the learned mapping from history to target becomes unstable around them and small changes can lead to very different predictions. In such cases, it is beneficial to introduce additional structural cues beyond the input history itself. In supervised forecasting datasets, these structural cues are naturally present in the third part of the chain: the post-target continuation segment. What happens after the target shows how such trajectories usually evolve—for example, whether an upward move tends to keep rising, flatten out, or reverse—and these regularities provide an additional inductive bias. Leveraging how similar training trajectories typically continue, this continuation effectively narrows the space of plausible futures for the current input window, reducing local ambiguity and stabilizing long-horizon predictions. Intuitively, exploiting continuation information makes the forecasting problem less like unconstrained extrapolation from the past alone and more like an interpolation-style task that is guided by how similar histories have actually evolved in the training data. This intuition is related to a classical result in function approximation (Cheney, 1998; DeVore & Lorentz, 1993), which shows that for any $L$-Lipschitz function, the worst-case error of interpolation (using both past and future samples) is strictly bounded above by that of extrapolation (using past samples only). We derive this error bound formally in Appendix A. Modern deep forecasters do not strictly satisfy the assumptions of this theorem, and in realistic forecasting problems true future values are never observed at test time. We therefore use the interpolation view only as a conceptual motivation: by distilling typical post-target continuation patterns from training chains into an auxiliary stream and fusing it with the backbone representation, we aim to partially recover the stability benefits of interpolation while still operating in a purely predictive setting where only past observations are available at test time.

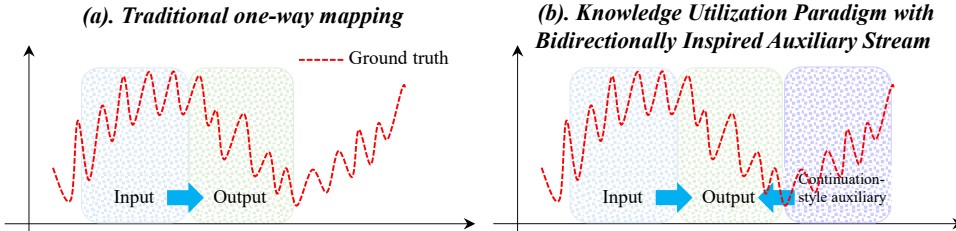

Figure 1: Comparison between (a) a traditional one-way mapping from history to target and (b) our knowledge utilization paradigm with bidirectionally inspired auxiliary Stream.

Building on this perspective, we propose the **Knowledge Utilization Paradigm with Bidirectionally Inspired Auxiliary Stream (KUP-BI)**, which augments the standard **history**-to-**target** pathway with a continuation-style auxiliary stream constructed from a train-only historical library $\mathcal{D}$. Rather than conditioning on any true future values at test time, KUP-BI computes a deterministic auxiliary feature $\mathbf{Z} = f(\mathbf{X}, \mathcal{D})$ that summarises how training trajectories with histories similar to the current input $\mathbf{X}$ have tended to continue beyond their targets. To construct $\mathbf{Z}$, we decompose the training data into "**history**–**target**–**post-target continuation**" chains and build a *train-only* library $\mathcal{D}$ in which each entry consists of a history window and the post-target continuation that follows its target. On each aligned history–continuation pair $(\mathbf{H}, \mathbf{F})$, we apply a simple ratio-style operator that encodes how the continuation changes relative to its history, and store the resulting history-to-continuation transformation together with its history in the library $\mathcal{D}$. Given a new input $\mathbf{X}$, we retrieve similar training histories from this library $\mathcal{D}$ (Han et al., 2025; Ning et al., 2025), aggregate their associated transformations via a temperature-controlled softmax weighting, and apply the aggregated transformation to $\mathbf{X}$ to obtain the continuation-style auxiliary feature $\mathbf{Z}$. In our framework, the current input and this continuation-style auxiliary feature are processed in two parallel branches and fused after feature extraction through a lightweight gating module, so that the backbone can exploit both the local information in $\mathbf{X}$ and the typical continuation patterns distilled from the library $\mathcal{D}$. Conceptually, this provides an additional, data-driven inductive bias that encourages forecasts to align with the kinds of post-target evolution commonly observed in the training data, rather than relying solely on parametric extrapolation from the current history. Figure 1 contrasts the conventional one-way mapping with this augmented scheme, highlighting how continuation-style auxiliary streams can act as structural guidance to stabilise predictions under complex dynamics (Kim et al. (2022)). It is important to note that KUP-BI is not bound to any specific mechanism for constructing this continuation-style auxiliary feature. As detailed in Appendix B, one can also obtain such a

feature from a predictor-based source, for example by letting the backbone produce a longer horizon in one shot and using the tail window as an auxiliary input. *The novelty of KUP-BI lies in explicitly introducing and exploiting continuation-style auxiliary features derived from training chains, and in demonstrating that injecting such features as a separate stream can consistently improve forecasting performance across diverse backbones.* Our contributions are as follows:

- We propose a new perspective for time-series forecasting that explicitly leverages the full "**history–target–post-target continuation**" chains present in the *training data*. Instead of relying solely on one-way extrapolation from history to target, we introduce continuation-style auxiliary features distilled from how training trajectories tend to evolve beyond their targets, and use them as a structural inductive bias for forecasting.

- We instantiate this idea with KUP-BI, a simple non-parametric framework that can be plugged into standard forecasting backbones.

- We demonstrate the generality of KUP-BI by incorporating it as a plug-in module into several state-of-the-art backbones across six standard benchmarks. In all cases, the continuation-style auxiliary stream consistently improves forecasting performance with small additional overhead, confirming the effectiveness of the proposed design.

## 2 RELATED WORK

### 2.1 TIME SERIES FORECASTING MODELS

In recent years, time-series forecasting techniques have developed rapidly to address increasingly complex prediction demands (Liu et al. (2024); Dai et al. (2024); Huang et al. (2025b)). Different neural network architectures show distinct advantages: convolution-based models (Wu et al. (2023); donghao & wang xue (2024)) can efficiently capture local temporal features, Transformer-based models (Nie et al. (2023); Zhou et al. (2022)) are effective at modeling long-range dependencies, and MLP-based models (Zeng et al. (2023a); Wang et al. (2024a)) are known for their simplicity and computational efficiency. With further research, large-scale pre-trained models (Jin et al. (2024); Niu et al. (2025)) have been explored for time-series forecasting, and these approaches demonstrate good performance in zero-shot and few-shot scenarios. Existing time-series forecasting models, despite their architectural diversity, still follow the traditional single-stream paradigm, relying solely on past-to-future mappings. This unidirectional design limits their ability to leverage future continuation streams as external knowledge, and may reduce robustness under distribution shifts (Kim et al. (2022); Liu et al. (2025)).

### 2.2 INFORMATION UTILIZATION IN FORECASTING

Considering information beyond the original input to enhance model performance has long been a hotspot in time-series forecasting. In the era when statistical forecasting dominated, ARIMAX (Williams (2001)) and SARIMAX (Vagropoulos et al. (2016))—which incorporate exogenous covariates—already outperformed their non-exogenous counterparts. In the deep learning stage, NBEATSx (Olivares et al. (2023)) and TiDE (Das et al. (2023)) directly take future exogenous features as inputs, while TimeXer (Wang et al. (2024b)) explicitly models the interaction between endogenous and exogenous variables within the Transformer framework and addresses issues such as lag and missing values in exogenous sequences. More recently, ExoLLM (Huang et al. (2025a)) has leveraged large language models (LLMs) to understand and model the multi-grained influence of exogenous variables, extracting textual knowledge to provide stronger generalization ability.

Another prevalent approach is to leverage the outputs of similar patterns to enhance the current prediction. For example, RAFT (Retrieval-Augmented Forecasting) (Han et al. (2025)) incorporates the outputs of similar patterns into the current prediction, effectively improving model accuracy. TS-RAG (Ning et al. (2025)) supplements the representation of the current input by retrieving the outputs corresponding to similar historical inputs, and feeds them together with the input sequence representation generated by the TSFM backbone into a Mixture-of-Experts (MoE) module to obtain richer representations. Such approaches are often implemented through retrieval mechanisms, owing to their efficiency and simplicity.

Both exogenous-variable methods and retrieval-based approaches enhance forecasting performance by injecting information beyond the raw inputs, which motivates us to ask how such additional structure can be exploited more systematically. From a broader perspective, retrieval-augmented approaches such as RAFT (Han et al. (2025)) and TS-RAG (Ning et al. (2025)), together with our KUP-BI framework, can also be viewed as non-parametric/meta-learning mechanisms: a train-only trajectory library plays the role of a support set, and test-time predictions are adapted by querying and recombining information from this set rather than by updating the backbone parameters (Hospedales et al., 2022; Vinyals et al., 2016). However, despite this shared reliance on extra information, existing exogenous-variable and retrieval-augmented methods still operate within the first two segments of the natural chain "*history* (model input), *target* (ground-truth output), *post-target continuation*": they enrich only the mapping between *history* and *target*, whereas KUP-BI explicitly brings the third segment into play by constructing a continuation-style auxiliary feature stream from the *post-target continuation* in the training data.

## 3 METHODOLOGY

**Problem Settings** For input sequences $\mathbf{X} = \{\boldsymbol{x}_i\}_{i=0}^{L-1} \in \mathbb{R}^{C \times L}$, where $C$ and $L$ denote the number of channels and the length of the look-back window, respectively. The goal of time-series forecasting is to learn a mapping from historical observations to the ground-truth target sequence, producing predictions $\hat{\mathbf{Y}} = \{\hat{\boldsymbol{y}}_i\}_{i=0}^{T-1} \in \mathbb{R}^{C \times T}$ that approximate the ground-truth series $\mathbf{Y} = \{\boldsymbol{y}_i\}_{i=0}^{T-1} \in \mathbb{R}^{C \times T}$, where $T$ is the forecast horizon. In particular, when $C = 1$ and $T = 1$, the task reduces to the most basic univariate prediction. In time-series forecasting, MAE and MSE are standard evaluation metrics, with lower values reflecting better predictive accuracy. Their definitions are as follows:

$$\mathcal{L}_{MAE} = \frac{1}{CT} \sum_{c=1}^{C} \sum_{i=0}^{T-1} |y_{c,i} - \hat{y}_{c,i}|, \qquad \mathcal{L}_{MSE} = \frac{1}{CT} \sum_{c=1}^{C} \sum_{i=0}^{T-1} (y_{c,i} - \hat{y}_{c,i})^2.$$

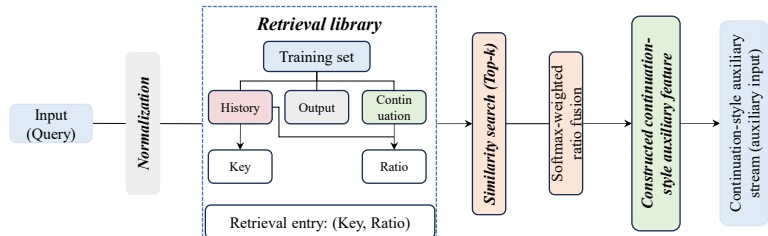

Figure 2: Overview of the proposed continuation-style auxiliary construction.

**Structure Overview (Continuation-Style Auxiliary Construction).** As shown in Figure 2, we instantiate the continuation-style auxiliary stream via a retrieval-based construction scheme. We represent the training set as a collection of "**history** (model input), **target** (ground-truth output), **post-target continuation**" chains and, for each chain, construct an aligned pair consisting of a history window and a post-target continuation window of the same length. On this aligned pair we derive a ratio matrix $\mathbf{R}$ that characterizes how the continuation changes relative to its history within that chain (*e.g.*, amplitude rescaling, seasonal strengthening or weakening, short-term carry-over).

Given a new input history, we retrieve the Top-$k$ chains whose histories are most similar to it, collect their corresponding ratio matrices, and aggregate them via a temperature-controlled softmax to obtain a fused ratio matrix. We then apply this fused ratio matrix to the current input to obtain a deterministic continuation-style auxiliary sequence, followed by a scale-normalization step. The current stream and this auxiliary stream are encoded into feature representations by separate encoders and then fused at the feature level by a lightweight gating module to form the final predictive representation.

*Note that the retrieval library is constructed once from the training set and then fixed; the same train-only library is used to generate continuation-style auxiliary streams for training, validation, and testing, ensuring a consistent setup without any leakage of validation or test futures into the library.*

**Assumption 1 (Conditional stationarity of continuation patterns).** For a given current history segment, we assume that its post-target continuation is similar to the post-target continuations of training history segments that are similar to it under the similarity measure $corr(\cdot, \cdot)$.

This is analogous to exchangeability assumptions commonly made in non-parametric regression and retrieval-based meta-learning Hospedales et al. (2022); Vinyals et al. (2016).

**Retrieval Library.** Given a multivariate time series, each training instance naturally contains a **history** segment $\mathbf{H} \in \mathbb{R}^{L \times C}$, its corresponding **target** segment $\mathbf{Y} \in \mathbb{R}^{T \times C}$ and a subsequent **post-target continuation** segment $\mathbf{F} \in \mathbb{R}^{L \times C}$. In our implementation, for each chain we take a history window and extract a post-target continuation window of the same length that follows the target in time. To obtain a simple description of how the continuation relates to its history, we compute a ratio-style representation between $\mathbf{H}$ and $\mathbf{F}$:

$$\mathbf{R} = \frac{\mathbf{F} - \mathbf{H}}{\mathbf{H} + \varepsilon \cdot \mathrm{sign}(\mathbf{H})}, \tag{1}$$

where the division is element-wise, $\mathrm{sign}(\cdot)$ is taken element-wise and $\varepsilon$ is a small stabiliser that prevents numerical instability near zero. This matrix $\mathbf{R}$ can be viewed as a heuristic relative-change descriptor that highlights how the post-target continuation differs from its history (for example, amplitude rescaling, seasonal strengthening or weakening), rather than as an optimal or unique statistical operator.

We deliberately adopt this closed-form ratio, instead of introducing an additional neural summariser over the library, for two reasons. First, it keeps the continuation construction strictly non-parametric and decoupled from the backbone, so that the same library can be reused across different backbones (or even non-neural forecasters) without retraining. Second, it avoids giving KUP-BI extra trainable capacity compared to the baselines, making it easier to attribute performance gains to how continuation-style information is utilised rather than to a larger model. More flexible learnable encoders over the library are orthogonal extensions and are left to future work.

Intuitively, for many short-horizon forecasting tasks the target segment $\mathbf{Y}$ is a local continuation of the history in both scale and shape. In such settings, the way a trajectory tends to change *relative to its recent past* is often more informative than its absolute level. The ratio-style representation $\mathbf{R}$ is designed precisely to capture these continuation patterns: it is approximately invariant to global rescaling and shifts of the raw series, while emphasising per-channel changes in amplitude, trend, and local oscillations. In other words, $\mathbf{R}$ serves as a compact surrogate for "how this kind of history typically keeps going".

We then build a retrieval library from the training set as a collection of history–ratio pairs:

$$\mathcal{D} = \{(\mathbf{H}_j, \mathbf{R}_j)\}_{j=1}^N, \tag{2}$$

where each entry corresponds to one training chain.

**Correlation-based candidate selection (channel-wise, offset-anchored).** Given a query (current input) window $\mathbf{X}_q \in \mathbb{R}^{L \times C}$, we first apply last-step offsetting (Han et al., 2025) to remove local level differences: $\tilde{\mathbf{X}}_q[t,:] = \mathbf{X}_q[t,:] - \mathbf{X}_q[L,:]$, $\tilde{\mathbf{H}}_j[t,:] = \mathbf{H}_j[t,:] - \mathbf{H}_j[L,:]$, $t = 1, \ldots, L$, where $\mathbf{X}_q[L,:]$ and $\mathbf{H}_j[L,:]$ denote the last time step of each window. We then compute *channel-wise* Pearson correlations (Benesty et al., 2009) between $\tilde{\mathbf{X}}_q^{(:,c)}$ and all $\tilde{\mathbf{H}}_j^{(:,c)}$: $\mathrm{corr}_{j,c} = \mathrm{Corr}(\tilde{\mathbf{X}}_q^{(:,c)}, \tilde{\mathbf{H}}_j^{(:,c)})$, $c = 1, \ldots, C$. For each channel $c$, we select its Top-$k$ neighbours according to the largest absolute correlations:

$$\mathcal{K}_c(\mathbf{X}_q) = \mathrm{Top}\text{-}k\left(\{|\mathrm{corr}_{j,c}|\}_j\right). \tag{3}$$

**Softmax-weighted ratio fusion (compact, per channel).** For each channel $c$, we aggregate the corresponding columns of the history-to-continuation ratio matrices from its Top-$k$ candidates $\mathcal{K}_c(\mathbf{X}_q)$ using a temperature-controlled softmax over the (absolute) correlations:

$$\hat{\mathbf{r}}^{(c)} = \sum_{j \in \mathcal{K}_c(\mathbf{X}_q)} \alpha_{j,c} \mathbf{R}_j^{(:,c)}, \qquad \alpha_{j,c} = \frac{\exp\left((|\mathrm{corr}_{j,c}| - m_c)/\tau\right)}{\sum_{\ell \in \mathcal{K}_c(\mathbf{X}_q)} \exp\left((|\mathrm{corr}_{\ell,c}| - m_c)/\tau\right)}, \tag{4}$$

where $\tau$ is a temperature parameter and $m_c = \max_{\ell \in \mathcal{K}_c(\mathbf{X}_q)} |\mathrm{corr}_{\ell,c}|$ is subtracted for numerical stability. Stacking $\{\hat{\mathbf{r}}^{(c)}\}_{c=1}^C$ yields the fused ratio matrix $\hat{\mathbf{R}}_q \in \mathbb{R}^{L \times C}$.

We then apply quantile–$\tanh$ clipping with the 90th percentile $R'_q = \mathcal{Q}_{0.9}(|\hat{\mathbf{R}}_q|)$:

$$\tilde{\mathbf{R}}_q = R'_q \cdot \tanh\!\big(\hat{\mathbf{R}}_q/R'_q\big), \tag{5}$$

where the division and $\tanh$ are applied element-wise. This softmax-weighted fusion emphasises candidates that are more strongly correlated with the query while retaining contributions from multiple neighbours, and the quantile-based $\tanh$ clipping provides a simple way to limit extreme ratio values and improve robustness across datasets.

**Continuation-style auxiliary generation.** We apply $\tilde{\mathbf{R}}_q$ to the current input to obtain a continuation-style auxiliary sequence: $\hat{\mathbf{F}}_q = \mathbf{X}_q + \tilde{\mathbf{R}}_q \odot \mathbf{X}_q$, where $\odot$ denotes element-wise multiplication. Equivalently, this can be written as $\hat{\mathbf{F}}_q = (\mathbf{1} + \tilde{\mathbf{R}}_q) \odot \mathbf{X}_q$, so $\tilde{\mathbf{R}}_q$ acts as a feature-wise multiplicative modulation of the current history rather than as a separate source of future values. To roughly align its scale with the historical stream, we further normalise $\hat{\mathbf{F}}_q$ to match the mean and (per-channel) standard deviation of $\mathbf{X}_q$:

$$\mathbf{Z} = \big(\hat{\mathbf{F}}_q - \boldsymbol{\mu}_{\hat{\mathbf{F}}_q}\big) \oslash \big(\boldsymbol{\sigma}_{\hat{\mathbf{F}}_q} + \varepsilon\big) \odot \big(\boldsymbol{\sigma}_{\mathbf{X}_q} + \varepsilon\big) + \boldsymbol{\mu}_{\mathbf{X}_q}, \tag{6}$$

where $\oslash$ denotes element-wise division, and $\boldsymbol{\mu}_{(\cdot)}$, $\boldsymbol{\sigma}_{(\cdot)}$ are the channel-wise mean and standard deviation computed over the time dimension.

This modulation step is loosely consistent with the ratio definition in (1): it uses the estimated ratio as a feature-wise scaling of the current history, but we do not treat it as an exact algebraic inverse of (1). Rather, the modulation and normalisation together form a simple practical heuristic that injects continuation-style information into the main stream while keeping the auxiliary branch on a comparable scale.

**Gated fusion with harmonic residual.** Let the intermediate features of the historical stream and the continuation-style auxiliary stream be $\mathbf{X}_{\mathrm{main}} = \mathrm{Fea}(\mathbf{X}_q)$ and $\mathbf{X}_{\mathrm{aux}} = \mathrm{Fea}(\mathbf{Z})$, respectively. We introduce a learnable gate parameter $\mathbf{g}$, which is passed through a sigmoid to obtain channel-wise weights $\boldsymbol{\gamma} = \sigma(\mathbf{g})$. The two streams are first combined via gated fusion:

$$\widetilde{\mathbf{X}} = \boldsymbol{\gamma} \odot \mathbf{X}_{\mathrm{main}} + (1 - \boldsymbol{\gamma}) \odot \mathbf{X}_{\mathrm{aux}}, \tag{7}$$

where $\odot$ denotes element-wise multiplication and the same $\boldsymbol{\gamma}$ is broadcast along the time dimension.

To further stabilise training and preserve the dominant role of the main (historical) stream, we apply a harmonic residual controlled by a coefficient $\alpha \in [0, 1]$: $\mathbf{X}' = \alpha \mathbf{X}_{\mathrm{main}} + (1 - \alpha) \widetilde{\mathbf{X}}$.

This design ensures that the final representation is always a convex combination of the two streams: $\boldsymbol{\gamma}$ adaptively balances the contribution of each channel within the fusion, while $\alpha$ controls the residual strength of the main stream so that the auxiliary branch acts as a modulation rather than a replacement. We regard this gating-and-residual fusion as a simple, stable heuristic for injecting continuation-style information into the backbone.

**Complexity Analysis.** The proposed KUP-BI paradigm comprises three components. First, a correlation-based retrieval stage builds a train-only library and computes correlations offline, caching per-channel candidate lists so this step does not affect training or inference latency. Second, an online continuation-style auxiliary computation (optionally) re-ranks cached candidates, applies a temperature-scaled softmax, fuses their ratios to form the auxiliary sequence and performs quantile-plus-tanh clipping with distribution alignment. Third, a lightweight gated fusion combines the historical stream and the auxiliary stream. Compared to typical backbones (*e.g.*, multi-head attention), these online operations are negligible in runtime and memory.

# 4 EXPERIMENTS

## 4.1 EXPERIMENT SETTING

**Datasets** We conduct experiments on six widely used public benchmarks covering electricity, health-care, and economics: the ETT family (ETTh1, ETTh2, ETTm1, ETTm2) (Zhou et al., 2021), ILI[1], and Exchange Rate (Lai et al., 2018). Detailed dataset statistics and preprocessing procedures are provided in Appendix C.

**Backbones** To evaluate the effectiveness and generality of the proposed KUP-BI, we instantiate it on four strong forecasting backbones with diverse architectures: the Transformer-based PatchTST (Nie et al., 2023), the MLP-based DLinear (Zeng et al., 2023a), the CNN-based TimesNet (Wu et al., 2023), and the hybrid dual-stream (MLP+CNN) xPatch (Stitsyuk & Choi, 2025). Backbone-specific details are summarized in Appendix D.

**Fusion point** We keep each backbone architecturally unchanged and only attach a lightweight dual-stream fusion block at a single feature interface where the historical stream is already formed. Specifically: (i) DLinear (Zeng et al., 2023a) and xPatch (Stitsyuk & Choi, 2025) both adopt a seasonal–trend two-stream decomposition, and we fuse the historical stream with the continuation-style auxiliary stream after the decomposition stage; (ii) PatchTST (Nie et al., 2023) fuses the historical stream and the auxiliary stream after patch projection and before the Transformer encoder; (iii) TimesNet (Wu et al., 2023) fuses them after the DataEmbedding layer and before the TimesBlocks. The gate is lightweight (per-channel/per-variable scalars with broadcasting), and the same fusion rule is used across all backbones.

**Setups** To ensure a fair comparison, we follow the default configurations used by the backbone papers. (1) **Prediction horizons.** For all datasets except ILI, we evaluate prediction lengths $\{96, 192, 336, 720\}$; for ILI we use $\{24, 36, 48, 60\}$. (2) **Input length.** The input sequence length follows the recommended configuration of each backbone model. (3) **Metrics.** We report mean squared error (MSE) and mean absolute error (MAE) (lower is better). (4) **Data splits.** For the ETT family (ETTh1/2, ETTm1/2), we adopt the standard 12/4/4-month train/validation/test split; for the other datasets we use a 7:1:2 chronological split.

**Implementation Details** All experiments are implemented in PyTorch (Paszke et al., 2019) and run on 1 NVIDIA RTX 4090 (24 GB) and 3 NVIDIA RTX 4080 (16 GB each) with driver 525.147.05 and CUDA 12.0. We use the Adam optimizer and MSE loss. For unstable models (TimesNet (Wu et al. (2023)) and xPatch (Stitsyuk & Choi (2025))), results are averaged over three independent runs. For xPatch, we conducted experiments by referring to the Settings in the file named *xPatch_fair* among the three scripts provided by the authors.

Table 1: Long-term multivariate forecasting results. Results are averaged from all prediction lengths. The better results are highlighted in **bold**. Full results are listed in **Appendix F.1**.

| Model | PatchTST | | | | DLinear | | | | TimesNet | | | | xPatch | | | |
|---|---|---|---|---|---|---|---|---|---|---|---|---|---|---|---|---|
| | Ori | | +KUP-BI | | Ori | | +KUP-BI | | Ori | | +KUP-BI | | Ori | | +KUP-BI | |
| Metric | MSE | MAE | MSE | MAE | MSE | MAE | MSE | MAE | MSE | MAE | MSE | MAE | MSE | MAE | MSE | MAE |
| ETTh1 | 0.419 | 0.432 | **0.409** | **0.425** | 0.445 | 0.454 | **0.425** | **0.437** | 0.472 | 0.463 | **0.453** | 0.453 | 0.444 | 0.438 | **0.409** | **0.422** |
| ETTh2 | 0.330 | 0.379 | **0.327** | **0.376** | 0.469 | 0.463 | **0.394** | **0.426** | 0.415 | 0.426 | **0.396** | **0.414** | 0.342 | 0.383 | **0.338** | **0.381** |
| ETTm1 | 0.353 | 0.382 | **0.350** | **0.379** | 0.359 | 0.381 | **0.358** | **0.380** | 0.415 | 0.418 | **0.410** | **0.417** | 0.352 | 0.372 | **0.350** | **0.372** |
| ETTm2 | 0.258 | 0.315 | **0.255** | **0.314** | 0.283 | 0.345 | **0.266** | **0.330** | 0.296 | 0.333 | **0.293** | **0.332** | 0.252 | 0.308 | **0.250** | **0.308** |
| ILI | 1.580 | 0.852 | **1.496** | **0.807** | 2.347 | 1.089 | **2.292** | **1.069** | 2.438 | 0.955 | **2.200** | **0.888** | 1.383 | 0.718 | **1.365** | **0.712** |
| Exchange | 0.385 | 0.418 | **0.367** | **0.411** | 0.369 | 0.418 | **0.312** | **0.389** | 0.415 | 0.440 | **0.397** | **0.432** | 0.364 | 0.403 | **0.359** | **0.402** |
| Improvement | 2.542% (MSE) | | 1.721% (MAE) | | 7.395% (MSE) | | 4.160% (MAE) | | 4.132% (MSE) | | 2.420% (MAE) | | 2.173% (MSE) | | 0.866% (MAE) | |

## 4.2 MAIN RESULTS

As shown in Table 1, KUP-BI consistently improves all four backbones (PatchTST, DLinear, Times-Net, xPatch) on six real-world datasets in terms of both MSE and MAE. Averaged over all prediction horizons and datasets, the relative MSE reductions are largest on DLinear (7.40%, MAE 4.16%), clear on TimesNet (4.13%, MAE 2.42%), and smaller but steady on PatchTST (2.54%, MAE 1.72%) and xPatch (2.17%, MAE 0.87%). We observe no degradation on any backbone–dataset combina-

---

[1]https://gis.cdc.gov/grasp/fluview/fluportaldashboard.html

tion. These results indicate that augmenting standard forecasters with an approximate future continuation stream is generally beneficial across diverse architectures.

**Why do gains differ across backbones?** We attribute the different gain patterns to three interacting factors: *(a) Capacity and headroom.* Lightweight models such as DLinear have limited capacity to capture complex seasonal and phase structure from history alone. The continuation-style auxiliary stream, distilled from training continuations, provides additional structural cues, leaving more headroom for improvement and yielding larger gains. *(b) Inductive-bias overlap.* TimesNet and xPatch already encode multi-periodicity and seasonality via CNN blocks or seasonal–trend two-stream decompositions. As a result, part of the structure carried by the auxiliary stream overlaps with what the backbone already models, so the marginal benefit is smaller than on DLinear but remains consistent. *(c) Fusion locus and dilution.* We fuse at the feature level. In DLinear/xPatch, fusion happens shallowly after seasonal–trend decomposition, directly steering the prediction heads. In deeper architectures such as PatchTST, the auxiliary features must traverse multiple layers and may be diluted by self-attention, leading to milder but stable improvements.

## 4.3 COMPARISON WITH SOTA RAFT

Table 2: KUP-BI vs. RAFT across four backbones on ETTh1/ETTh2 and four horizons (MSE/MAE↓). Best results in bold; "Avg" averages across horizons.

| Model | PatchTST | | | | DLinear | | | | TimesNet | | | | xPatch | | | |
|---|---|---|---|---|---|---|---|---|---|---|---|---|---|---|---|---|
| | KUP-BI | | RAFT | | KUP-BI | | RAFT | | KUP-BI | | RAFT | | KUP-BI | | RAFT | |
| ETTh1 | MSE | MAE | MSE | MAE | MSE | MAE | MSE | MAE | MSE | MAE | MSE | MAE | MSE | MAE | MSE | MAE |
| 96 | **0.364** | **0.391** | 0.366 | 0.392 | **0.372** | **0.394** | 0.402 | 0.415 | **0.387** | **0.412** | 0.416 | 0.429 | **0.359** | **0.389** | 0.361 | 0.391 |
| 192 | **0.404** | **0.415** | 0.405 | 0.415 | **0.406** | **0.415** | 0.465 | 0.460 | **0.446** | **0.447** | 0.464 | 0.458 | **0.405** | **0.414** | 0.410 | 0.418 |
| 336 | 0.434 | 0.440 | **0.422** | **0.430** | **0.443** | **0.443** | 0.499 | 0.492 | **0.487** | **0.470** | 0.494 | 0.474 | **0.423** | **0.428** | 0.436 | 0.434 |
| 720 | 0.432 | 0.456 | **0.432** | **0.455** | **0.479** | **0.495** | 0.587 | 0.565 | **0.493** | **0.483** | 0.546 | 0.515 | **0.450** | **0.458** | 0.455 | 0.463 |
| Avg | 0.409 | 0.425 | **0.406** | **0.423** | **0.425** | **0.437** | 0.488 | 0.483 | **0.453** | **0.453** | 0.480 | 0.469 | **0.409** | **0.422** | 0.415 | 0.426 |
| ETTh2 | MSE | MAE | MSE | MAE | MSE | MAE | MSE | MAE | MSE | MAE | MSE | MAE | MSE | MAE | MSE | MAE |
| 96 | **0.272** | **0.333** | 0.278 | 0.337 | **0.282** | **0.347** | 0.441 | 0.436 | **0.319** | **0.362** | 0.320 | 0.368 | **0.274** | **0.333** | 0.285 | 0.341 |
| 192 | **0.335** | **0.376** | 0.340 | 0.380 | **0.344** | **0.393** | 0.527 | 0.486 | **0.395** | **0.407** | 0.413 | 0.422 | **0.337** | **0.375** | 0.347 | 0.381 |
| 336 | **0.326** | **0.377** | 0.335 | 0.388 | **0.400** | **0.435** | 0.539 | 0.498 | **0.435** | **0.439** | 0.453 | 0.455 | **0.362** | **0.398** | 0.365 | 0.402 |
| 720 | **0.375** | **0.419** | 0.382 | 0.424 | **0.550** | **0.528** | 0.662 | 0.560 | **0.435** | **0.449** | 0.508 | 0.488 | **0.381** | **0.420** | 0.394 | 0.434 |
| Avg | **0.327** | **0.376** | 0.334 | 0.382 | **0.394** | **0.426** | 0.542 | 0.495 | **0.396** | **0.414** | 0.424 | 0.433 | **0.338** | **0.381** | 0.348 | 0.390 |

We compare KUP-BI with the plug-in version of RAFT (Han et al., 2025) under strictly matched conditions: identical backbones (PatchTST, DLinear, TimesNet, xPatch), the same training/validation/test splits, optimisers, learning-rate schedules, and training budgets. For both methods, the retrieval library is constructed once from the training range only and then fixed; RAFT is given access to exactly the same library and similarity metric as KUP-BI, and neither method ever uses validation or test trajectories inside the library. No extra training data are reserved exclusively for KUP-BI. In particular, the KUP-BI gating/fusion head is trained only on the same supervised pairs (history, target) as the plain backbones and RAFT plug-ins, so that any performance gains can be attributed to how the shared library is utilised rather than to additional data or capacity. RAFT otherwise follows its original design: for each query history, it retrieves the Top-$k$ neighbours from the library and aggregates their targets into a retrieval-enhanced prediction, whereas KUP-BI constructs a continuation-style auxiliary stream from the same neighbours and fuses it with the backbone features at the representation level.

As shown in Table 2, KUP-BI consistently matches or outperforms RAFT across ETTh1 and ETTh2 on three backbones (DLinear, TimesNet, xPatch), and is competitive on PatchTST. On the simpler DLinear and xPatch models, KUP-BI yields clearly lower MSE/MAE on both datasets (*e.g.*, on ETTh1 and ETTh2, KUP-BI improves the average MSE over RAFT for all four horizons), while on the stronger PatchTST backbone KUP-BI is slightly better on ETTh2 and comparable on ETTh1. Overall, these results indicate that explicitly constructing and fusing a future continuation stream is at least as effective as, and often more effective than, reusing retrieved past targets in the RAFT-style convex combination, especially on lightweight backbones with more headroom for improvement.

## 4.4 ABLATION ON KUP-BI COMPONENTS (DLINEAR BACKBONE)

We ablate the three main components of KUP-BI on ETTm2, ILI and Exchange using DLinear as the backbone. All runs share the same data split, optimiser, schedule and hyperparameters; we only remove the specified component: 1) **w/o distribution alignment:** remove the per-channel

mean/variance matching between the constructed continuation-style auxiliary sequence and the current input history. 2) **w/o ratio:** retrieve Top-$k$ training chains whose histories match the current input and directly use their aligned post-target continuation segments (after mean/variance alignment) as the auxiliary sequence, without applying the ratio-style operator. 3) **w/o $\alpha$:** drop the residual anchoring term and rely on the learned gate alone for fusion. The results are reported in Table 3, and additional experiments are provided in **Appendix F.3**.

Table 3: Ablation study of the components of KUP-BI on the ETTm2, ILI and Exchange datasets using DLinear as a backbone. The best results are highlighted in **bold**, and the second-best results are highlighted in underline. Full results are listed in **Appendix F.3**.

| Dataset | | ETTm2 | | | | | ILI | | | | | Exchange | | | | |
|---|---|---|---|---|---|---|---|---|---|---|---|---|---|---|---|---|
| Forecast length | | 96 | 192 | 336 | 720 | Avg | 24 | 36 | 48 | 60 | Avg | 96 | 192 | 336 | 720 | Avg |
| KUP-BI | MSE | 0.166 | 0.222 | **0.299** | 0.377 | **0.266** | **2.224** | **2.225** | 2.266 | 2.453 | 2.292 | 0.086 | **0.153** | 0.355 | **0.653** | **0.312** |
| | MAE | **0.258** | 0.300 | 0.364 | **0.400** | **0.330** | **1.036** | **1.057** | 1.060 | 1.121 | 1.069 | **0.207** | **0.285** | 0.437 | 0.628 | **0.389** |
| w/o distribution | MSE | 0.280 | 0.358 | 0.384 | 0.455 | 0.369 | 2.238 | 2.267 | 2.288 | 2.507 | 2.325 | 0.345 | 0.796 | 2.013 | 1.314 | 1.117 |
| | MAE | 0.338 | 0.388 | 0.401 | 0.443 | 0.392 | 1.041 | 1.064 | 1.062 | 1.125 | 1.073 | 0.378 | 0.584 | 0.934 | 0.881 | 0.694 |
| w/o ratio | MSE | 0.165 | 0.221 | 0.299 | 0.377 | 0.266 | 2.229 | 2.237 | **2.253** | **2.430** | 2.287 | **0.085** | 0.156 | **0.353** | 0.669 | 0.316 |
| | MAE | 0.258 | **0.299** | 0.363 | 0.400 | 0.330 | 1.038 | 1.061 | **1.056** | **1.114** | **1.068** | 0.207 | 0.287 | **0.434** | 0.635 | 0.391 |
| w/o $\alpha$ | MSE | 0.166 | 0.222 | 0.302 | 0.379 | 0.267 | 2.482 | 2.359 | 2.482 | 2.658 | 2.495 | 0.087 | 0.154 | 0.356 | 0.840 | 0.359 |
| | MAE | 0.259 | 0.300 | 0.365 | 0.401 | 0.332 | 1.089 | 1.075 | 1.119 | 1.178 | 1.115 | 0.208 | 0.286 | 0.438 | 0.687 | 0.405 |

**Distribution alignment is critical.** Removing distribution alignment causes the largest and most systematic degradation across all datasets and horizons (see Table 3). The effect is especially pronounced on ETTm2 and Exchange, where the scales of the input history and the auxiliary sequence differ substantially; without alignment, the gate sees mismatched distributions and either under-uses or over-trusts the auxiliary stream, hurting both MSE and MAE.

**Residual anchoring ($\alpha$) stabilises fusion.** Removing $\alpha$ (gate-only fusion) yields consistent MAE increases and frequent MSE regressions. The residual path keeps the main historical stream dominant and lets the model inject the auxiliary stream only when it is consistent with the history, reducing over-fitting to noisy or partially mismatched auxiliary features.

**Ratio versus direct continuation segments.** On ETTm2 and ILI, replacing the ratio-style transformation with direct post-target continuation segments from the library (under the same mean/variance alignment) yields results comparable to KUP-BI; by contrast, on Exchange, KUP-BI with the ratio-style operator maintains a clear advantage. Intuitively, the ratio encodes a relative history-to-continuation gain and is applied to the current history, preserving the sample's shape and phase while primarily modulating amplitude, which can make it more robust to amplitude mismatch and phase drift. Taken together, these results suggest that ratio-based continuation-style auxiliaries can be advantageous, especially on datasets with stronger scale shifts, while also confirming that the overall benefit of KUP-BI does not hinge on a single specific operator and that simpler alternatives can already be competitive on some datasets.

## 4.5 Hyperparameter Sensitivity

We study three hyperparameters in our method: the number of retrieved items (Top-$k$), the softmax temperature ($\tau$), and the gating coefficient ($\alpha$). Under a fixed retrieval library, data split, and random seed, we vary one hyperparameter at a time while holding the other two at the setting indicated in each subfigure, and report MSE for horizons 96/192/336/720. Figure 3 summarizes results on ETTh1 (top row) and ETTh2 (bottom row) with PatchTST as the backbone.

**1)** Top-$k$. Across both ETTh1 ($\tau = 1.0$, $\alpha = 0.9$) and ETTh2 ($\tau = 10.0$, $\alpha = 0.95$), the curves are nearly flat as $k$ increases ($2\rightarrow10$), indicating low sensitivity. Very large $k$ tends to dilute structure without gains. Recommendation: keep a small-to-moderate $k$ (3–6). **2)** $\alpha$. ETTh1 ($\tau = 1.0$, Top-$k$=6): performance improves mildly but consistently as $\alpha$ increases ($0.55\rightarrow0.90$), with longer horizons (336/720) benefiting most. A larger $\alpha$ behaves like a conservative controller that down-weights mismatched future continuation streams. ETTh2 ($\tau = 10.0$, Top-$k$=8): curves are essentially flat, that is, very robust to $\alpha$. **3)** $\tau$. ETTh1 ($\alpha = 0.9$, Top-$k$=6): almost invariant from $\tau = 0.01$ to 10 $\rightarrow$ low sensitivity. ETTh2 ($\alpha = 0.95$, Top-$k$=8): when $\tau$ reaches 0.05, as $\tau$ increases, the model's results remain unchanged, indicating that the model is not sensitive to changes in $\tau$.

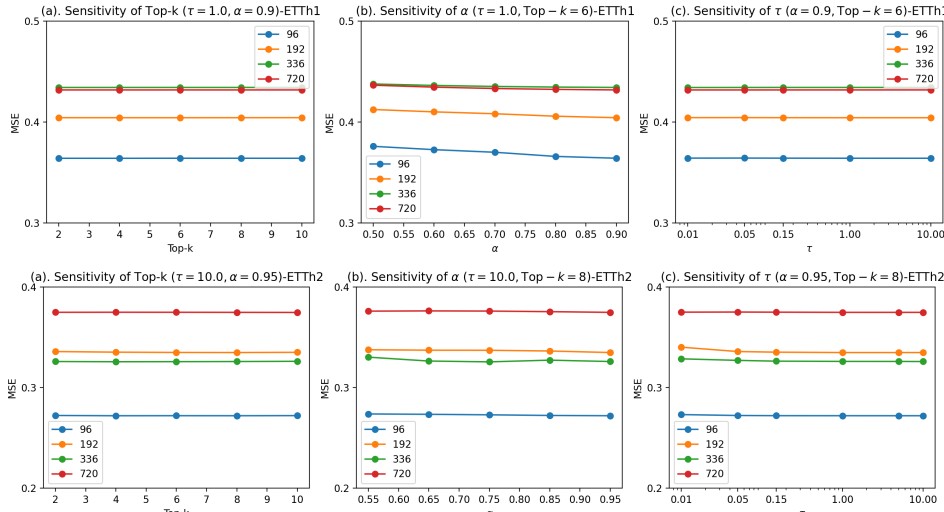

Figure 3: Hyperparameter sensitivity on ETTh1/2 with PatchTST. We vary one hyperparameter while fixing the other two at the best setting and plot MSE for horizons 96/192/336/720.

For PatchTST the three knobs are broadly insensitive, especially Top-$k$ and $\tau$. Using a slightly conservative gate (larger $\alpha$) and a moderately sharp temperature (small $\tau$) yields stable performance across horizons.

## 5 CONCLUSION AND FUTURE WORK

We presented KUP-BI, a knowledge-utiliation paradigm that augments time-series forecasting with a continuation-style auxiliary stream. The auxiliary stream is constructed in a non-parametric way from train-only "history–target–post-target continuation" chains via simple history-to-continuation ratio-style transformations, and is fused with the current input stream through a lightweight, feature-level gated module. Across multiple datasets and backbones, KUP-BI yields small but consistent error reductions with modest computational overhead, suggesting that explicitly leveraging post-target continuations from the training data can provide a useful structural bias for forecasting.

From a statistical perspective, KUP-BI can be viewed as a non-parametric, library-based estimator built on top of a backbone model. Providing a rigorous analysis of its statistical properties—including the precise exchangeability conditions under which Assumption 1 holds, its asymptotic consistency and the bias–variance trade-offs induced by the library size, Top-$k$ and temperature—is an important next step. A distilled continuation-based linear variant, outlined in Appendix **??**, provides a simple setting in which continuation information can be analysed using standard tools from non-parametric regression; we leave a full statistical treatment and empirical study of this variant to future work. Finally, we believe the insights from KUP-BI—explicitly constructing and utilising continuation-style auxiliary features derived from training chains, rather than relying solely on one-way extrapolation from history to target—can inform the design of retrieval-augmented and LLM-based time-series forecasters at larger scales.

## 6 ETHICS STATEMENT

This study is limited to time series forecasting and therefore entails no potential ethical risks.

## 7 REPRODUCIBILITY STATEMENT

In the main text, all details are clearly described through formulas. The experimental details, including dataset descriptions, models, and others, are provided in the appendix. The code will be released after the paper is accepted.

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

## USE OF GENERATIVE AI.

We used a large language model (LLM) to assist with (i) language editing (grammar/wording/typos), (ii) minor rewriting for clarity, and (iii) refactoring non-novel code utilities (*e.g.*, plotting scripts) that do not affect scientific claims. All technical ideas, experiment designs, analyses, and conclusions are authored by the authors. We verified every LLM-assisted snippet and did not rely on the LLM for data collection, experimental results, or figure generation beyond cosmetic formatting.

## A  PROOF OF ERROR BOUND FOR INTERPOLATION VS. EXTRAPOLATION

**Definition 1** (*L*-Lipschitz Continuity). A function $f : [t_0, t_1] \to \mathbb{R}$ is said to be $L$-Lipschitz continuous if for any $s, t \in [t_0, t_1]$, the following inequality holds:
$$|f(s) - f(t)| \leq L|s - t|.$$

**Theorem 1** (Error Bound of Interpolation with Future Continuation). Let $f : [t_0, t_1] \to \mathbb{R}$ be an $L$-Lipschitz function. For any point $t^* \in (t_0, t_1)$, the following holds:

1): Extrapolation error (using only the historical value at $t_0$): The worst-case error bound is
$$|f(t^*) - f(t_0)| \leq L|t^* - t_0| = L\Delta.$$

2): Interpolation error (using the historical value at $t_0$ and a future continuation $\hat{f}(t_1)$ with error bounded by $\varepsilon$):

Let $\hat{f}(t_1)$ be an approximation of $f(t_1)$ such that $\left|\hat{f}(t_1) - f(t_1)\right| \leq \varepsilon$. The linear interpolation estimate is defined as:
$$\hat{f} = \alpha f(t_0) + (1 - \alpha)\tilde{f}(t_1), \alpha = \frac{t_1 - t^*}{t_1 - t_0}.$$

Then the error of this interpolation is bounded by:
$$\left|\hat{f} - f(t^*)\right| \leq L\frac{t_1 - t_0}{2} + (1 - \alpha)\varepsilon.$$

3): Interpolation is always superior to extrapolation.

*Proof.* The error bound for extrapolation follows directly from Definition 1.

For the interpolation error, we begin by decomposing the error term:
$$
\begin{aligned}
\left|\hat{f} - f(t^*)\right| &= \left|\alpha f(t_0) + (1 - \alpha)\hat{f}(t_1) - f(t^*)\right| \\
&= \left|\alpha(f(t_0) - f(t^*)) + (1 - \alpha)\left(\hat{f}(t_1) - f(t^*)\right)\right| \\
&\leq \alpha|f(t_0) - f(t^*)| + (1 - \alpha)\left|\hat{f}(t_1) - f(t^*)\right|
\end{aligned}
$$

Since $f$ is $L$-Lipschitz, we have $|f(t_0) - f(t^*)| \leq L|t_0 - t^*| = L\Delta$ , where $\Delta = t^* - t_0$ .

For the second term, we apply the triangle inequality and the Lipschitz condition again:

$$\left|\hat{f}(t_1) - f(t^*)\right| \leq \left|\hat{f}(t_1) - f(t_1)\right| + |f(t_1) - f(t^*)| \leq \varepsilon + L|t_1 - t^*| = \varepsilon + L\Delta', \Delta' = t_1 - t^*.$$

Substituting these inequalities back yields:

$$\left|\hat{f} - f(t^*)\right| \leq \alpha L\Delta + (1 - \alpha)(\varepsilon + L\Delta') = L(\alpha\Delta + (1 - \alpha)\Delta') + (1 - \alpha)\varepsilon.$$

We now simplify the term $\alpha\Delta + (1 - \alpha)\Delta'$ . Recalling that $\alpha = \frac{t_1 - t^*}{t_1 - t_0}$ and $1 - \alpha = \frac{t^* - t_0}{t_1 - t_0}$ we have:

$$\alpha\Delta + (1 - \alpha)\Delta' = \frac{t_1 - t^*}{t_1 - t_0}\Delta + \frac{t^* - t_0}{t_1 - t_0}\Delta' = \frac{(t_1 - t^*)\Delta + (t^* - t_0)\Delta'}{t_1 - t_0} = \frac{2\Delta\Delta'}{t_1 - t_0}.$$

By the inequality of arithmetic and geometric means (AM-GM), we have:

$$\Delta\Delta' \leq \left(\frac{\Delta + \Delta'}{2}\right)^2 = \left(\frac{t_1 - t_0}{2}\right)^2.$$

Therefore,

$$\alpha\Delta + (1 - \alpha)\Delta' \leq \left(\frac{t_1 - t_0}{2}\right)^2 \frac{2}{t_1 - t_0} = \frac{t_1 - t_0}{2}.$$

Substituting this back, we obtain the general bound:

$$\left|\hat{f} - f(t^*)\right| \leq L\left(\frac{t_1 - t_0}{2}\right)^2 \frac{2}{t_1 - t_0} + (1 - \alpha)\varepsilon.$$

Since $1 - \alpha = \frac{t^* - t_0}{t_1 - t_0} = \frac{\Delta}{t_1 - t_0} \leq 1$ , it follows that $(1 - \alpha)\varepsilon \leq \varepsilon$, namely,

$$\left|\hat{f} - f(t^*)\right| \leq L\frac{t_1 - t_0}{2} + \varepsilon.$$

When $\varepsilon = 0$ or $\varepsilon$ is small, $L\frac{t_1 - t_0}{2} < L|t_1 - t_0|$ .

Therefore, interpolation is always superior to extrapolation.

This completes the proof.

## B PREDICTOR-BASED ALTERNATIVE AUXILIARY CONSTRUCTION (CONCEPT ONLY)

**Idea.** The two-stream interface in KUP-BI accepts any proposal that has the same shape as the history $\mathbf{H}$. Beyond retrieval, a *predictor-based continuation-style auxiliary* can be obtained by making the backbone output a longer horizon in a single forward pass (for example, input length 336 and output length 336+336) and taking the tail window $\hat{\mathbf{Y}}_{336:672}$ as an auxiliary sequence with the same length as the history. This illustrates that our two-stream design is, by construction, agnostic to the specific source used to construct the auxiliary sequence, as long as it is a deterministic function of the current input (and fixed model parameters).

**Why we do not include it as a comparison experiment.**

(1) *Modelling complexity.* One-shot longer-horizon prediction typically requires modifying or at least fine-tuning the prediction head to 336+336 and carefully handling timestamps; even a zero-training rolling variant introduces notable runtime variance.

(2) *Quality sensitivity.* The tail window of a long-horizon forecast tends to be noisier than the first horizon; without additional phase correction or regularisation, its usefulness as an auxiliary sequence depends on careful gating tuning, which would substantially expand the hyperparameter budget.

Table 4: Detailed dataset descriptions.

| Dataset | Information | Datset Size | Length | Channel | Frequency |
|---------|-------------|-------------|--------|---------|-----------|
| ETTh1 ETTh2 | Energy | (8545, 2881, 2881) | 17420 | 7 | 1 hour |
| ETTm1 ETTm2 | | (34465, 11521, 11521) | 69680 | | 15 min |
| ILI | Healthcare | (617, 74, 170) | 966 | 7 | 1 Week |
| Exchange | Finance | (5120, 665, 1422) | 7588 | 8 | 1 day |
| Weather | Weather | (36792, 5271, 10540) | 52696 | 21 | 10 min |

(3) *Fairness and scope.* Our contribution targets the *form and usage* of continuation-style auxiliary features (for example, simple ratio-style transformations plus gated residual fusion), rather than an exhaustive comparison over all possible auxiliary sources. In this paper we therefore focus on a single, simple retrieval-based instantiation to keep the experimental scope controlled.

**Takeaway.** The existence of this predictor-based alternative supports the claim that our two-stream design does not rely on a particular retrieval mechanism: any reasonable continuation-style auxiliary sequence (retrieval-based, predictor-based or hybrid), viewed as a deterministic function of the input and a fixed library or model, can in principle be plugged into the same ratio-style transformation and gated-residual fusion machinery without changing the core framework.

## C  DATASET DESCRIPTIONS

In this paper, we consider six datasets to verify the effectiveness of the method proposed in this paper. Their details are as follows.

**ETT (Electricity Transformer Temperature)** (Zhou et al. (2021)): The ETT dataset consists of two subsets with an hourly granularity ("h") and two subsets with a 15-minute granularity ("m"). These datasets contain electricity transformer data collected from two different counties between July 2016 and July 2018, with seven recorded features. The suffixes "1" and "2" distinguish the two regions where the data were collected.

**Exchange** (Lai et al. (2018)): It contains the daily exchange rates of eight countries from 1990 to 2016, including Australia, the United Kingdom, Canada, Switzerland, China, Japan, New Zealand and Singapore.

**ILI** [2]: The ILI dataset reports the proportion of patients diagnosed with influenza-like illness relative to the total number of patients. It provides weekly records collected by the U.S. Centers for Disease Control and Prevention (CDC) spanning the years 2002 to 2021.

The data processing and train–validation–test splitting schemes described in earlier literature (Nie et al. (2023); Zeng et al. (2023a); Wu et al. (2023)) are followed in this paper. For the ETT dataset, the split ratio is 6:2:2, while for the other datasets it is 7:1:2. Details of the datasets are provided in Table 4.

## D  BACKBONE MODELS

In this paper, we select one representative state-of-the-art model from each of the three most classic architectures as our backbone: Transformer-based PatchTST (Nie et al. (2023)), MLP-based DLinear (Zeng et al. (2023a)), CNN-based TimesNet (Wu et al. (2023)), and a hybrid dual-stream (MLP+CNN) non-Transformer model xPatch (Stitsyuk & Choi (2025)). We largely follow the official implementations of these backbones. Specifically, for PatchTST, we fix the batch size and apply Bayesian search to tune the other hyperparameters. For the other two models, our search space covers the most important parameters (*e.g.*, learning rate, batch_size, d_model, etc.).

**PatchTST** [3]: A model based on patch learning that adopts a channel-independent approach for multivariate time-series modeling.

---

[2] https://gis.cdc.gov/grasp/fluview/fluportaldashboard.html
[3] https://github.com/yuqinie98/PatchTST

**DLinear**[4]: A streamlined model whose learning process involves only decomposition and linear layers.

**TimesNet**[5]: It converts one-dimensional time series into two-dimensional representations to model multi-periodic patterns and leverages CNNs to capture dependencies both across and within periods.

**xPatch** [6]: A dual-stream CNN–MLP architecture that applies exponential seasonal-trend decomposition, patching, and channel-independence to capture both linear and nonlinear patterns in multivariate time-series forecasting.

# E    CONCATENATION BASELINE VS. GATED FUSION

We compare two ways of utilising the continuation-style auxiliary stream, keeping all other settings fixed: (1) **Concatenation.** The constructed continuation-style auxiliary sequence is appended after the current input along the time axis, and the resulting longer sequence is fed into the backbone as a single stream (with extended time encoding). (2) **Fusion (ours).** The historical stream and the continuation-style auxiliary stream are encoded separately, and their feature representations are fused by the lightweight gated module described in Section 3.

Table 5 shows that Fusion consistently outperforms Concatenation across four backbones and all six datasets in both MSE and MAE, with only rare near-ties (*e.g.*, TimesNet on ETTm2). The gap is particularly pronounced on simpler models such as DLinear (*e.g.*, ETTh2: MSE 0.394 vs. 0.464; ILI: MSE 2.292 vs. 2.512), and remains steady on PatchTST, TimesNet, and xPatch. These results indicate that the way in which the future continuation is integrated into the model matters: naive concatenation is not sufficient.

Table 5: Fusion (ours) vs. Concatenation across ETTh1/ETTh2/ETTm1/ETTm2, ILI, and Exchange with four backbones (PatchTST, DLinear, TimesNet, xPatch). Results are averaged from all prediction lengths. The better results are highlighted in **bold**. Full results are listed in **Appendix F.2**.

| Model | PatchTST+KUP-BI | | | | DLinear+KUP-BI | | | | TimesNet+KUP-BI | | | | xPatch+KUP-BI | | | |
|---|---|---|---|---|---|---|---|---|---|---|---|---|---|---|---|---|
| | **Fusion** | | Concatenation | | **Fusion** | | Concatenation | | **Fusion** | | Concatenation | | **Fusion** | | Concatenation | |
| Metric | MSE | MAE | MSE | MAE | MSE | MAE | MSE | MAE | MSE | MAE | MSE | MAE | MSE | MAE | MSE | MAE |
| ETTh1 | **0.409** | **0.425** | 0.427 | 0.435 | **0.425** | **0.437** | 0.466 | 0.468 | **0.453** | **0.453** | 0.501 | 0.475 | **0.409** | **0.422** | 0.412 | 0.429 |
| ETTh2 | **0.327** | **0.376** | 0.331 | 0.381 | **0.394** | **0.426** | 0.464 | 0.463 | **0.396** | **0.414** | 0.404 | 0.418 | **0.338** | **0.381** | 0.350 | 0.389 |
| ETTm1 | **0.350** | **0.379** | 0.394 | 0.407 | **0.358** | **0.380** | 0.377 | 0.391 | **0.410** | 0.417 | 0.413 | **0.416** | **0.350** | **0.372** | 0.396 | 0.404 |
| ETTm2 | **0.255** | **0.314** | 0.263 | 0.321 | **0.266** | **0.330** | 0.303 | 0.361 | **0.293** | **0.332** | 0.294 | 0.332 | **0.250** | **0.308** | 0.296 | 0.341 |
| ILI | **1.496** | **0.807** | 1.714 | 0.885 | **2.292** | **1.069** | 2.512 | 1.147 | **2.200** | **0.888** | 2.814 | 1.085 | **1.365** | **0.712** | 1.854 | 0.874 |
| Exchange | **0.367** | **0.411** | 0.395 | 0.424 | **0.312** | **0.389** | 0.392 | 0.434 | **0.397** | **0.432** | 0.400 | 0.433 | **0.359** | **0.402** | 0.379 | 0.410 |

# F    FULL RESULTS

## F.1    FULL FORECASTING RESULTS

Due to page limits, we summarize the full multivariate forecasting results in Table 6. Under the same training/evaluation budget, augmenting each backbone with KUP-BI yields consistent improvements across all datasets (and most forecast horizons), as evidenced by lower MSE/MAE.

## F.2 FULL PER-HORIZON RESULTS: FUSION (OURS) VS. CONCATENATION ACROSS BACKBONES AND DATASETS

Table 7 summarizes how the two strategies (Concatenation and Fusion) affect model performance. Across ETTh1/2, ETTm1/2, ILI, and Exchange, Fusion (ours) consistently outperforms Concatenation for all four backbones, with only rare near ties.

---

[4]https://github.com/cure-lab/LTSF-Linear

[5]https://github.com/thuml/Time-Series-Library

[6]https://github.com/stitsyuk/xPatch/tree/main

Table 6: Full results of the long-term forecasting task. Avg means the average results from all four prediction lengths. The better results are highlighted in **bold**.

| Model | | PatchTST | | | | DLinear | | | | TimesNet | | | | xPatch | | | |
|---|---|---|---|---|---|---|---|---|---|---|---|---|---|---|---|---|---|
| | | Ori | | +KUP-BI | | Ori | | +KUP-BI | | Ori | | +KUP-BI | | Ori | | +KUP-BI | |
| Metric | | MSE | MAE | MSE | MAE | MSE | MAE | MSE | MAE | MSE | MAE | MSE | MAE | MSE | MAE | MSE | MAE |
| ETTh1 | 96 | 0.382 | 0.405 | **0.364** | **0.391** | 0.384 | 0.405 | **0.372** | **0.394** | 0.396 | 0.417 | **0.387** | **0.412** | 0.369 | 0.397 | **0.359** | **0.389** |
| | 192 | 0.414 | 0.421 | **0.404** | **0.415** | 0.443 | 0.450 | **0.406** | **0.415** | 0.466 | 0.459 | **0.446** | **0.447** | 0.424 | 0.426 | **0.405** | **0.414** |
| | 336 | **0.431** | **0.436** | 0.434 | 0.440 | 0.447 | 0.448 | **0.443** | **0.443** | 0.506 | 0.477 | **0.487** | **0.470** | 0.450 | 0.437 | **0.423** | **0.428** |
| | 720 | 0.449 | 0.466 | **0.432** | **0.456** | 0.504 | 0.515 | **0.479** | **0.495** | 0.521 | 0.497 | **0.493** | **0.483** | 0.533 | 0.494 | **0.450** | **0.458** |
| | Avg | 0.419 | 0.432 | **0.409** | **0.425** | 0.445 | 0.454 | **0.425** | **0.437** | 0.472 | 0.463 | **0.453** | **0.453** | 0.444 | 0.438 | **0.409** | **0.422** |
| ETTh2 | 96 | 0.275 | 0.337 | **0.272** | **0.333** | 0.290 | 0.353 | **0.282** | **0.347** | 0.336 | 0.372 | **0.319** | **0.362** | 0.275 | 0.333 | **0.274** | **0.333** |
| | 192 | 0.338 | 0.378 | **0.335** | **0.376** | 0.388 | 0.422 | **0.344** | **0.393** | 0.423 | 0.426 | **0.395** | **0.407** | 0.337 | 0.375 | **0.337** | **0.375** |
| | 336 | 0.329 | 0.379 | **0.326** | **0.377** | 0.463 | 0.473 | **0.400** | **0.435** | 0.444 | 0.444 | **0.435** | **0.439** | 0.365 | 0.398 | **0.362** | **0.398** |
| | 720 | 0.379 | 0.422 | **0.375** | **0.419** | 0.733 | 0.606 | **0.550** | **0.528** | 0.457 | 0.464 | **0.435** | **0.449** | 0.391 | 0.426 | **0.381** | **0.420** |
| | Avg | 0.330 | 0.379 | **0.327** | **0.376** | 0.469 | 0.463 | **0.394** | **0.426** | 0.415 | 0.426 | **0.396** | **0.414** | 0.342 | 0.383 | **0.338** | **0.381** |
| ETTm1 | 96 | **0.292** | **0.343** | 0.295 | 0.345 | 0.301 | 0.345 | **0.300** | **0.344** | 0.338 | 0.375 | 0.339 | 0.378 | 0.289 | 0.332 | 0.289 | 0.333 |
| | 192 | 0.336 | 0.371 | **0.329** | **0.368** | 0.336 | 0.366 | **0.335** | **0.366** | 0.403 | 0.408 | **0.385** | **0.400** | 0.330 | 0.357 | 0.331 | 0.358 |
| | 336 | 0.366 | 0.392 | **0.362** | **0.387** | 0.372 | 0.389 | 0.373 | 0.390 | 0.421 | 0.424 | **0.414** | **0.423** | 0.364 | **0.381** | 0.363 | 0.382 |
| | 720 | 0.418 | 0.424 | **0.413** | **0.417** | 0.427 | 0.423 | **0.425** | **0.420** | **0.498** | **0.464** | 0.504 | 0.467 | 0.426 | 0.417 | **0.419** | **0.414** |
| | Avg | 0.353 | 0.382 | **0.350** | **0.379** | 0.359 | 0.381 | **0.358** | **0.380** | 0.415 | 0.418 | **0.410** | **0.417** | 0.352 | 0.372 | **0.350** | 0.372 |
| ETTm2 | 96 | 0.165 | **0.255** | **0.165** | 0.256 | 0.172 | 0.267 | **0.166** | **0.258** | 0.188 | 0.266 | **0.185** | **0.266** | **0.159** | **0.245** | 0.160 | 0.246 |
| | 192 | 0.220 | **0.292** | **0.220** | 0.293 | 0.237 | 0.314 | **0.222** | **0.300** | 0.257 | 0.312 | **0.251** | **0.306** | 0.217 | **0.286** | **0.217** | 0.288 |
| | 336 | 0.278 | 0.329 | **0.274** | **0.327** | **0.295** | **0.359** | 0.299 | 0.364 | 0.322 | 0.350 | **0.320** | **0.349** | 0.275 | 0.325 | **0.274** | **0.325** |
| | 720 | 0.368 | 0.385 | **0.362** | **0.381** | 0.427 | 0.439 | **0.377** | **0.400** | 0.405 | 0.418 | 0.406 | **0.406** | 0.357 | 0.377 | **0.348** | **0.374** |
| | Avg | 0.258 | 0.315 | **0.255** | **0.314** | 0.283 | 0.345 | **0.266** | **0.330** | 0.296 | 0.333 | **0.293** | **0.332** | 0.252 | 0.308 | **0.250** | **0.308** |
| ILI | 24 | 1.584 | 0.840 | **1.409** | **0.781** | 2.280 | 1.061 | **2.224** | **1.036** | 2.662 | 0.974 | **2.531** | **0.884** | 1.334 | 0.699 | **1.323** | **0.693** |
| | 36 | 1.442 | 0.831 | **1.341** | **0.765** | 2.235 | 1.059 | **2.225** | **1.057** | 2.756 | 1.010 | **2.116** | **0.904** | 1.329 | 0.683 | **1.300** | **0.675** |
| | 48 | 1.685 | 0.853 | **1.596** | **0.825** | 2.298 | 1.079 | **2.266** | **1.060** | 2.299 | 0.922 | **2.246** | **0.892** | **1.358** | **0.706** | 1.373 | 0.711 |
| | 60 | **1.608** | **0.886** | 1.636 | 0.860 | 2.573 | 1.157 | **2.453** | **1.121** | 2.035 | 0.912 | **1.907** | **0.872** | 1.512 | 0.785 | **1.463** | **0.770** |
| | Avg | 1.580 | 0.852 | **1.496** | **0.807** | 2.347 | 1.089 | **2.292** | **1.069** | 2.438 | 0.955 | **2.200** | **0.888** | 1.383 | 0.718 | **1.365** | **0.712** |
| Exchange | 96 | 0.097 | 0.217 | **0.089** | **0.211** | **0.085** | 0.209 | 0.086 | **0.207** | **0.108** | 0.236 | 0.110 | 0.238 | **0.081** | **0.197** | 0.085 | 0.202 |
| | 192 | **0.193** | **0.313** | 0.199 | 0.322 | 0.162 | 0.296 | **0.153** | **0.285** | 0.213 | 0.335 | **0.203** | **0.325** | **0.175** | **0.296** | 0.182 | 0.302 |
| | 336 | 0.364 | 0.439 | **0.338** | **0.421** | **0.333** | 0.441 | 0.355 | **0.437** | 0.367 | 0.439 | **0.358** | **0.435** | **0.343** | **0.421** | 0.344 | 0.422 |
| | 720 | 0.885 | 0.703 | **0.841** | **0.690** | 0.898 | 0.725 | **0.653** | **0.628** | 0.971 | 0.751 | **0.916** | **0.731** | 0.857 | 0.698 | **0.825** | **0.684** |
| | Avg | 0.385 | 0.418 | **0.367** | **0.411** | 0.369 | 0.418 | **0.312** | **0.389** | 0.415 | 0.440 | **0.397** | **0.432** | 0.364 | 0.403 | **0.359** | **0.402** |

## F.3 FULL ABLATION RESULTS OF KUP-BI COMPONENTS (W/O RATIO, W/O DISTRIBUTION ALIGNMENT, W/O $\alpha$)

Tables 8–11 report ablations of KUP-BI on four backbones. Because input-level normalization in PatchTST/TimesNet/xPatch already neutralizes scale mismatches, the distribution-alignment component is only meaningful for DLinear; for the other three backbones we therefore report two ablations: w/o ratio and w/o $\alpha$. Overall, alpha and ratio are indispensable because the overall performance of the w/o ratio and w/o $\alpha$ methods is not as good as that of the original KUP-BI.

## G TRAINING-TIME COST (MS/BATCH) ON ETTH1/2

We measured training time per batch (including forward and backward passes) on ETTh1/2 for four backbones (PatchTST, DLinear, TimesNet, xPatch) and four prediction horizons (96/192/336/720), using the same batch size and a single GPU. The detailed numbers are reported in Table 12.

Averaged over the four horizons on ETTh1, we obtain: 1) PatchTST: $11.6 \rightarrow 15.4$ ms/batch (+3.8 ms); 2) DLinear: $2.28 \rightarrow 5.08$ ms/batch (+2.8 ms); 3) TimesNet: $36.8 \rightarrow 77.4$ ms/batch (+40.7 ms); 4) xPatch: $11.0 \rightarrow 11.0$ ms/batch (essentially unchanged).

On ETTh2, we observe similar trends: PatchTST and xPatch incur at most about 0-20% extra time ($\leq$3-4 ms per batch), DLinear shows a larger relative factor but the absolute overhead remains small (around 1–3 ms/batch), and TimesNet shows about a two-fold increase in ms/batch under our current unoptimized retrieval implementation.

Overall, these results show that for typical patch-based backbones (PatchTST, xPatch), KUP-BI behaves as a lightweight plug-in with only a few additional milliseconds per batch. For very small or very heavy models (DLinear, TimesNet), the relative overhead can be higher, which is mainly due to our current prototype of exact correlation-based retrieval over a large library. This cost is not inherent to KUP-BI itself and can be further reduced with standard engineering optimizations.

Table 7: Fusion (ours) vs. Concatenation across ETTh1/ETTh2/ETTm1/ETTm2, ILI, and Exchange with four backbones (PatchTST, DLinear, TimesNet, xPatch). Avg means the average results from all four prediction lengths. The better results are highlighted in **bold**.

| Model | | PatchTST+KUP-BI | | | | DLinear+KUP-BI | | | | TimesNet+KUP-BI | | | | xPatch+KUP-BI | | | |
|---|---|---|---|---|---|---|---|---|---|---|---|---|---|---|---|---|---|
| | | Fusion | | Concatenation | | Fusion | | Concatenation | | Fusion | | Concatenation | | Fusion | | Concatenation | |
| Metric | | MSE | MAE | MSE | MAE | MSE | MAE | MSE | MAE | MSE | MAE | MSE | MAE | MSE | MAE | MSE | MAE |
| ETTh1 | 96 | **0.364** | **0.391** | 0.389 | 0.408 | **0.372** | **0.394** | 0.413 | 0.425 | **0.387** | **0.412** | 0.408 | 0.424 | **0.359** | **0.389** | 0.366 | 0.397 |
| | 192 | **0.404** | **0.415** | 0.418 | 0.422 | **0.406** | **0.415** | 0.453 | 0.453 | **0.446** | **0.447** | 0.513 | 0.476 | **0.405** | **0.414** | 0.412 | 0.422 |
| | 336 | **0.434** | **0.440** | 0.457 | 0.453 | **0.443** | **0.443** | 0.467 | 0.463 | **0.487** | **0.470** | 0.537 | 0.491 | **0.423** | **0.428** | 0.424 | 0.431 |
| | 720 | **0.432** | **0.456** | 0.445 | 0.457 | **0.479** | **0.495** | 0.532 | 0.531 | **0.493** | **0.483** | 0.544 | 0.508 | 0.450 | **0.458** | **0.446** | 0.466 |
| | Avg | **0.409** | **0.425** | 0.427 | 0.435 | **0.425** | **0.437** | 0.466 | 0.468 | **0.453** | **0.453** | 0.501 | 0.475 | **0.409** | **0.422** | 0.412 | 0.429 |
| ETTh2 | 96 | **0.272** | **0.333** | 0.280 | 0.341 | **0.282** | **0.347** | 0.311 | 0.371 | **0.319** | **0.362** | 0.322 | 0.362 | **0.274** | **0.333** | 0.277 | 0.338 |
| | 192 | **0.335** | **0.376** | 0.336 | 0.377 | **0.344** | **0.393** | 0.388 | 0.418 | **0.395** | **0.407** | 0.407 | 0.413 | **0.337** | **0.375** | 0.354 | 0.384 |
| | 336 | **0.326** | **0.377** | 0.326 | 0.382 | **0.400** | **0.435** | 0.467 | 0.469 | 0.435 | 0.439 | **0.432** | **0.438** | **0.362** | **0.398** | 0.374 | 0.407 |
| | 720 | **0.375** | **0.419** | 0.383 | 0.423 | **0.550** | **0.528** | 0.690 | 0.593 | **0.435** | **0.449** | 0.455 | 0.459 | **0.381** | **0.420** | 0.392 | 0.427 |
| | Avg | **0.328** | **0.378** | 0.331 | 0.381 | **0.394** | **0.426** | 0.464 | 0.463 | **0.396** | **0.414** | 0.404 | 0.418 | **0.338** | **0.381** | 0.350 | 0.389 |
| ETTm1 | 96 | **0.295** | **0.345** | 0.349 | 0.384 | **0.300** | **0.344** | 0.324 | 0.360 | **0.339** | 0.378 | 0.342 | **0.375** | **0.289** | **0.333** | 0.342 | 0.375 |
| | 192 | **0.329** | **0.368** | 0.380 | 0.403 | **0.335** | **0.366** | 0.356 | 0.380 | **0.385** | **0.400** | 0.390 | 0.402 | **0.331** | **0.358** | 0.366 | 0.389 |
| | 336 | **0.362** | **0.387** | 0.399 | 0.407 | **0.373** | **0.390** | 0.385 | 0.395 | **0.414** | **0.423** | 0.424 | 0.425 | **0.363** | **0.382** | 0.402 | 0.407 |
| | 720 | **0.413** | **0.417** | 0.448 | 0.435 | **0.425** | **0.420** | 0.443 | 0.429 | 0.504 | 0.467 | **0.496** | **0.461** | **0.419** | **0.414** | 0.472 | 0.446 |
| | Avg | **0.350** | **0.379** | 0.394 | 0.407 | **0.358** | **0.380** | 0.377 | 0.391 | **0.410** | 0.417 | 0.413 | **0.416** | **0.350** | **0.372** | 0.396 | 0.404 |
| ETTm2 | 96 | **0.165** | **0.256** | 0.173 | 0.265 | **0.166** | **0.258** | 0.182 | 0.280 | **0.185** | **0.266** | 0.190 | 0.267 | **0.160** | **0.246** | 0.184 | 0.272 |
| | 192 | **0.220** | **0.293** | 0.227 | 0.299 | **0.222** | **0.300** | 0.251 | 0.330 | 0.251 | **0.306** | **0.250** | 0.307 | **0.217** | **0.288** | 0.256 | 0.319 |
| | 336 | **0.274** | **0.327** | 0.283 | 0.335 | **0.299** | 0.364 | 0.301 | **0.360** | **0.320** | **0.349** | 0.321 | 0.350 | **0.274** | **0.325** | 0.320 | 0.359 |
| | 720 | **0.362** | **0.381** | 0.368 | 0.386 | **0.377** | **0.400** | 0.478 | 0.474 | 0.418 | **0.406** | **0.416** | 0.406 | **0.348** | **0.374** | 0.425 | 0.413 |
| | Avg | **0.255** | **0.314** | 0.263 | 0.321 | **0.266** | **0.330** | 0.303 | 0.361 | **0.293** | **0.332** | 0.294 | 0.332 | **0.250** | **0.308** | 0.296 | 0.341 |
| ILI | 24 | **1.409** | **0.781** | 1.687 | 0.870 | **2.224** | **1.036** | 2.647 | 1.179 | **2.531** | **0.884** | 2.955 | 1.123 | **1.323** | **0.693** | 2.207 | 0.991 |
| | 36 | **1.341** | **0.765** | 1.536 | 0.846 | **2.225** | **1.057** | 2.476 | 1.147 | **2.116** | **0.904** | 3.139 | 1.126 | **1.300** | **0.675** | 1.497 | 0.757 |
| | 48 | **1.596** | **0.825** | 1.815 | 0.911 | **2.266** | **1.060** | 2.372 | 1.116 | **2.246** | **0.892** | 2.693 | 1.071 | **1.373** | **0.711** | 1.374 | 0.742 |
| | 60 | **1.636** | **0.860** | 1.817 | 0.912 | **2.453** | **1.121** | 2.551 | 1.146 | **1.907** | **0.872** | 2.469 | 1.021 | **1.463** | **0.770** | 2.339 | 1.006 |
| | Avg | **1.496** | **0.807** | 1.714 | 0.885 | **2.292** | **1.069** | 2.512 | 1.147 | **2.200** | **0.888** | 2.814 | 1.085 | **1.365** | **0.712** | 1.854 | 0.874 |
| Exchange | 96 | **0.089** | **0.211** | 0.094 | 0.217 | **0.086** | **0.207** | 0.098 | 0.226 | **0.110** | **0.238** | 0.113 | 0.240 | 0.085 | **0.202** | **0.084** | 0.202 |
| | 192 | **0.199** | **0.322** | 0.205 | 0.324 | **0.153** | **0.285** | 0.195 | 0.315 | **0.203** | 0.325 | 0.203 | **0.324** | 0.182 | 0.302 | **0.179** | **0.301** |
| | 336 | **0.338** | **0.421** | 0.381 | 0.447 | **0.355** | **0.437** | 0.435 | 0.497 | **0.358** | **0.435** | 0.366 | 0.441 | 0.344 | 0.422 | **0.321** | **0.409** |
| | 720 | **0.841** | **0.690** | 0.901 | 0.707 | **0.653** | **0.628** | 0.840 | 0.697 | **0.916** | 0.731 | 0.918 | **0.728** | **0.825** | **0.684** | 0.933 | 0.729 |
| | Avg | **0.367** | **0.411** | 0.395 | 0.424 | **0.312** | **0.389** | 0.392 | 0.434 | **0.397** | **0.432** | 0.400 | 0.433 | **0.359** | **0.402** | 0.379 | 0.410 |

Table 8: Ablation study of the components of KUP-BI on the ETTh2, ETTm2, ILI and Exchange datasets using DLinear as a backbone. The best results are highlighted in **bold**, and the second-best results are highlighted in underline.

| Model | | DLinear + KUP-BI | | w/o distribution | | w/o ratio | | w/o α | |
|---|---|---|---|---|---|---|---|---|---|
| Metric | | MSE | MAE | MSE | MAE | MSE | MAE | MSE | MAE |
| ETTh2 | 96 | **0.282** | **0.347** | 0.662 | 0.528 | 0.285 | 0.348 | 0.282 | 0.347 |
| | 192 | **0.344** | **0.393** | 0.907 | 0.608 | 0.350 | 0.396 | 0.345 | 0.394 |
| | 336 | 0.400 | 0.435 | 1.125 | 0.692 | **0.395** | **0.429** | 0.399 | 0.435 |
| | 720 | **0.550** | **0.528** | 1.263 | 0.763 | 0.553 | 0.529 | 0.550 | 0.528 |
| | Avg | **0.394** | **0.426** | 0.989 | 0.648 | 0.396 | 0.426 | 0.394 | 0.426 |
| ETTm2 | 96 | 0.166 | **0.258** | 0.280 | 0.338 | **0.165** | 0.258 | 0.166 | 0.259 |
| | 192 | 0.222 | **0.300** | 0.358 | 0.388 | **0.221** | 0.299 | 0.222 | 0.300 |
| | 336 | **0.299** | 0.364 | 0.384 | 0.401 | 0.299 | **0.363** | 0.302 | 0.365 |
| | 720 | **0.377** | 0.400 | 0.455 | 0.443 | 0.377 | **0.400** | 0.379 | 0.401 |
| | Avg | **0.266** | **0.330** | 0.369 | 0.392 | 0.266 | 0.330 | 0.267 | 0.332 |
| ILI | 24 | **2.224** | **1.036** | 2.238 | 1.041 | 2.229 | 1.038 | 2.482 | 1.089 |
| | 36 | **2.225** | **1.057** | 2.267 | 1.064 | 2.237 | 1.061 | 2.359 | 1.075 |
| | 48 | 2.266 | 1.060 | 2.288 | 1.062 | **2.253** | **1.056** | 2.482 | 1.119 |
| | 60 | 2.453 | 1.121 | 2.507 | 1.125 | **2.430** | **1.114** | 2.658 | 1.178 |
| | Avg | 2.292 | 1.069 | 2.325 | 1.073 | **2.287** | **1.068** | 2.495 | 1.115 |
| Exchange | 96 | 0.086 | **0.207** | 0.345 | 0.378 | **0.085** | 0.207 | 0.087 | 0.208 |
| | 192 | **0.153** | **0.285** | 0.796 | 0.584 | 0.156 | 0.287 | 0.154 | 0.286 |
| | 336 | 0.355 | 0.437 | 2.013 | 0.934 | **0.353** | **0.434** | 0.356 | 0.438 |
| | 720 | **0.653** | **0.628** | 1.314 | 0.881 | 0.669 | 0.635 | 0.840 | 0.687 |
| | Avg | **0.312** | **0.389** | 1.117 | 0.694 | 0.316 | 0.391 | 0.359 | 0.405 |

# H    RELATIONSHIP BETWEEN AUXILIARY QUALITY AND FORECASTING GAINS

In this section, we evaluated the impact of the quality of the auxiliary quality on the forecasting gains. For each sample, we align the constructed continuation-style auxiliary segment (length $L$, channels $C$) with its corresponding ground-truth post-target segment (the $L$ steps immediately after the target window $T$), flatten both into vectors, and compute the Pearson correlation coefficient $r$ and the mean squared error of the auxiliary ($MSE_{\text{aux}}$).

Table 13 reports both the forecasting metrics (MSE and MAE) and the auxiliary-quality metrics (mean $|r|$, $MSE_{\text{aux}}$) on ETTh1/2 and ETTm1/2. To ensure fairness and reproducibility, all hyper-

Table 9: Ablation study of the components of KUP-BI on the ETTh2, ETTm2, ILI and Exchange datasets using PatchTST as a backbone. The best results are highlighted in **bold**, and the second-best results are highlighted in underline.

| Model | | PatchTST + KUP-BI | | w/o ratio | | w/o $\alpha$ | |
|---|---|---|---|---|---|---|---|
| Metric | | MSE | MAE | MSE | MAE | MSE | MAE |
| ETTh2 | 96 | **0.272** | **0.333** | 0.274 | 0.336 | 0.278 | 0.340 |
| | 192 | **0.335** | **0.376** | 0.348 | 0.382 | 0.344 | 0.386 |
| | 336 | **0.326** | **0.377** | 0.331 | 0.387 | 0.333 | 0.387 |
| | 720 | **0.375** | **0.419** | 0.385 | 0.427 | 0.381 | 0.422 |
| | Avg | **0.327** | **0.376** | 0.335 | 0.383 | 0.334 | 0.384 |
| ETTm2 | 96 | **0.165** | **0.256** | 0.166 | 0.256 | 0.168 | 0.258 |
| | 192 | **0.220** | **0.293** | 0.222 | 0.294 | 0.231 | 0.302 |
| | 336 | **0.274** | **0.327** | 0.274 | 0.327 | 0.285 | 0.334 |
| | 720 | **0.362** | **0.381** | 0.363 | 0.381 | 0.377 | 0.390 |
| | Avg | **0.255** | **0.314** | 0.257 | 0.314 | 0.265 | 0.321 |
| ILI | 24 | 1.409 | 0.781 | **1.392** | **0.772** | 2.174 | 1.055 |
| | 36 | 1.341 | 0.765 | **1.321** | **0.758** | 1.497 | 0.819 |
| | 48 | **1.596** | **0.825** | 1.683 | 0.836 | 1.673 | 0.879 |
| | 60 | 1.636 | 0.860 | **1.630** | **0.859** | 2.066 | 0.986 |
| | Avg | **1.496** | **0.807** | 1.507 | 0.807 | 1.852 | 0.935 |
| Exchange | 96 | 0.089 | 0.211 | **0.088** | **0.209** | 0.091 | 0.213 |
| | 192 | 0.199 | 0.322 | **0.196** | **0.317** | 0.202 | 0.324 |
| | 336 | 0.338 | 0.421 | **0.334** | **0.418** | 0.351 | 0.431 |
| | 720 | **0.841** | **0.690** | 0.900 | 0.708 | 0.845 | 0.691 |
| | Avg | **0.367** | **0.411** | 0.380 | 0.413 | 0.372 | 0.415 |

Table 10: Ablation study of the components of KUP-BI on the ETTh2, ETTm2, ILI and Exchange datasets using TimesNet as a backbone. The best results are highlighted in **bold**, and the second-best results are highlighted in underline.

| Model | | TimesNet + KUP-BI | | w/o ratio | | w/o $\alpha$ | |
|---|---|---|---|---|---|---|---|
| Metric | | MSE | MAE | MSE | MAE | MSE | MAE |
| ETTh2 | 96 | **0.319** | **0.362** | 0.339 | 0.373 | 0.322 | 0.364 |
| | 192 | **0.395** | **0.407** | 0.397 | 0.408 | 0.414 | 0.414 |
| | 336 | 0.435 | **0.439** | 0.436 | 0.440 | **0.432** | 0.439 |
| | 720 | 0.435 | **0.449** | 0.435 | 0.450 | 0.434 | 0.450 |
| | Avg | **0.396** | **0.414** | 0.402 | 0.418 | 0.401 | 0.417 |
| ETTm2 | 96 | **0.185** | **0.266** | 0.189 | 0.268 | 0.188 | 0.269 |
| | 192 | **0.251** | **0.306** | 0.251 | 0.307 | 0.251 | 0.308 |
| | 336 | 0.320 | 0.349 | **0.312** | **0.345** | 0.318 | 0.349 |
| | 720 | **0.418** | **0.406** | 0.418 | 0.407 | 0.431 | 0.415 |
| | Avg | **0.293** | **0.332** | 0.293 | 0.332 | 0.297 | 0.335 |
| ILI | 24 | **2.531** | **0.884** | 2.569 | 0.920 | 3.346 | 1.150 |
| | 36 | **2.116** | **0.904** | 2.236 | 0.929 | 2.521 | 1.040 |
| | 48 | 2.246 | 0.892 | **2.140** | **0.881** | 2.207 | 0.962 |
| | 60 | **1.907** | **0.872** | 1.956 | 0.883 | 2.199 | 0.975 |
| | Avg | **2.200** | **0.888** | 2.225 | 0.903 | 2.568 | 1.032 |
| Exchange | 96 | 0.110 | 0.238 | **0.109** | **0.236** | 0.116 | 0.244 |
| | 192 | **0.203** | 0.325 | 0.203 | **0.327** | 0.207 | 0.327 |
| | 336 | 0.358 | **0.435** | 0.358 | 0.436 | 0.363 | 0.439 |
| | 720 | 0.916 | 0.731 | 0.918 | 0.731 | **0.912** | **0.727** |
| | Avg | **0.397** | **0.432** | 0.397 | 0.432 | 0.400 | 0.434 |

parameters (Top-$k$, $\tau$, $L$, $T$, etc.) follow the main experiments, and the retrieval library is built only from the training set and kept frozen during evaluation to avoid data leakage; the ground-truth post-target segments are used here solely as a diagnostic signal to assess the quality of the auxiliary sequence, not as an input to the model.

From Table 13, we observe a clear trend: KUP-BI's gains are positively associated with the quality of the continuation-style auxiliary sequence. Within datasets of the same type and temporal granularity, a higher-quality auxiliary (larger $\mathrm{mean}\,|r|$, smaller $MSE_{\mathrm{aux}}$) tends to yield larger improvements. For example, between ETTh1 and ETTh2, ETTh2 has both higher correlation and lower MSE than ETTh1; accordingly, KUP-BI achieves larger error reductions on ETTh2. Overall, the closer the auxiliary sequence is to the true post-target continuation, the more reliably KUP-BI reduces forecasting error; in more irregular or high-noise settings, the auxiliary becomes less informative, and the benefit correspondingly diminishes.

# I   ALIGNMENT ASSUMPTION AND POSSIBLE EXTENSIONS

Our current ratio operator assumes elementwise alignment between the history and its post-target continuation within each training chain: we pair an equal-length history window and post-target continuation window and compute their ratio at the same time index. The fused ratio is then applied

Table 11: Ablation study of the components of KUP-BI on the ETTh2, ETTm2, ILI and Exchange datasets using xPatch as a backbone. The best results are highlighted in **bold**, and the second-best results are highlighted in underline.

| Model | | xPatch + KUP-BI | | w/o ratio | | w/o $\alpha$ | |
|---|---|---|---|---|---|---|---|
| Metric | | MSE | MAE | MSE | MAE | MSE | MAE |
| ETTh2 | 96 | **0.274** | **0.333** | 0.275 | 0.335 | 0.290 | 0.352 |
| | 192 | **0.337** | **0.375** | 0.348 | 0.381 | 0.357 | 0.396 |
| | 336 | **0.362** | **0.398** | 0.364 | 0.399 | 0.366 | 0.406 |
| | 720 | **0.381** | **0.420** | 0.384 | 0.424 | 0.407 | 0.439 |
| | Avg | **0.338** | **0.381** | 0.343 | 0.385 | 0.355 | 0.398 |
| ETTm2 | 96 | **0.160** | **0.246** | 0.160 | 0.246 | 0.175 | 0.265 |
| | 192 | **0.217** | **0.288** | 0.217 | 0.288 | 0.235 | 0.308 |
| | 336 | **0.274** | **0.325** | 0.275 | 0.325 | 0.290 | 0.344 |
| | 720 | **0.348** | **0.374** | 0.349 | 0.374 | 0.372 | 0.393 |
| | Avg | **0.250** | **0.308** | 0.250 | 0.308 | 0.268 | 0.327 |
| ILI | 24 | 1.323 | 0.693 | **1.320** | **0.694** | 1.592 | 0.783 |
| | 36 | **1.300** | **0.675** | 1.322 | 0.682 | 1.547 | 0.780 |
| | 48 | 1.373 | 0.711 | **1.356** | **0.705** | 1.437 | 0.741 |
| | 60 | **1.463** | **0.770** | 1.535 | 0.787 | 1.842 | 0.887 |
| | Avg | **1.365** | **0.712** | 1.383 | 0.717 | 1.604 | 0.798 |
| Exchange | 96 | 0.085 | 0.202 | **0.084** | **0.201** | 0.093 | 0.214 |
| | 192 | 0.182 | 0.302 | **0.181** | **0.301** | 0.193 | 0.310 |
| | 336 | 0.344 | 0.422 | 0.344 | 0.423 | 0.370 | 0.441 |
| | 720 | **0.825** | **0.684** | 0.827 | 0.685 | 0.883 | 0.710 |
| | Avg | **0.359** | **0.402** | 0.359 | 0.402 | 0.385 | 0.418 |

elementwise to the current input. This simple design keeps the computation cheap and transparent, but it is sensitive to phase shifts: when two trajectories have similar shapes but are shifted in time, the elementwise ratio can become noisy and reduce the quality of the constructed continuation-style auxiliary sequence.

A natural extension is to explicitly align the retrieved segments before forming the ratio. For example, one could (i) search over a small lag window and choose the phase shift that maximizes cross-correlation, then align the retrieved history–post-target continuation pair to this lag; (ii) adopt a lightweight DTW-style alignment to handle local time misalignment before computing the ratio; or (iii) define the ratio in the frequency domain (e.g., based on amplitude spectra), which is inherently more robust to phase differences. In all of these cases, the overall KUP-BI framework would remain unchanged and only the local definition of the ratio operator would become phase-aware. We leave a systematic exploration of such phase-aware variants, and their impact on the quality of the continuation-style auxiliary sequence, as an interesting direction for future work.

Table 12: Training-time cost (ms/batch) on ETTh1/2 for four backbones.

| Model | | PatchTST | | DLinear | | TimesNet | | xPatch | |
|---|---|---|---|---|---|---|---|---|---|
| | | Ori | KUP-BI | Ori | KUP-BI | Ori | KUP-BI | Ori | KUP-BI |
| ETTh1 | 96 | 11.15 | 14.79 | 2.16 | 6.23 | 27.97 | 36.76 | 9.57 | 9.30 |
| | 192 | 11.06 | 15.12 | 2.29 | 4.21 | 30.62 | 51.56 | 10.52 | 9.69 |
| | 336 | 11.74 | 15.82 | 2.14 | 4.77 | 35.00 | 70.39 | 11.51 | 11.64 |
| | 720 | 12.47 | 15.95 | 2.53 | 5.12 | 53.54 | 151.01 | 12.36 | 13.17 |
| ETTh2 | 96 | 11.06 | 10.09 | 1.92 | 2.76 | 29.25 | 51.67 | 6.35 | 7.75 |
| | 192 | 10.92 | 10.14 | 2.03 | 2.87 | 33.29 | 61.38 | 6.51 | 8.12 |
| | 336 | 11.47 | 10.28 | 2.06 | 2.90 | 38.66 | 75.20 | 6.64 | 7.88 |
| | 720 | 12.21 | 10.64 | 2.22 | 3.01 | 65.41 | 165.50 | 7.25 | 8.42 |

Table 13: Effect of Retrieved Continuation-style Auxiliary on Forecasting Accuracy (DLinear, Test Set).

| Dataset | DLinear | | DLinear+KUP-BI | | Improvement | | $mean\ |r|$ | $MSE_{pr}$ |
|---|---|---|---|---|---|---|---|---|
| | MSE | MAE | MSE | MAE | MSE | MAE | | |
| ETTh1 | 0.445 | 0.454 | 0.425 | 0.437 | 4.403% | 3.848% | 0.588 | 0.803 |
| ETTh2 | 0.469 | 0.463 | 0.394 | 0.426 | 15.887% | 8.096% | 0.883 | 0.342 |
| ETTm1 | 0.359 | 0.381 | 0.358 | 0.380 | 0.181% | 0.176% | 0.471 | 1.252 |
| ETTm2 | 0.283 | 0.345 | 0.266 | 0.330 | 5.930% | 4.179% | 0.926 | 0.222 |

## J  A SIMPLE CONTINUATION-BASED LINEAR MODEL

In this appendix we sketch a mathematically simpler variant of our continuation-based idea that removes the deep backbone and keeps only a basic continuation summary and a linear predictor. This toy model is not part of our main experiments, but it illustrates in a clean way how information from post-target continuations can be incorporated into a coherent estimator.

**Setup.** We consider a univariate time series $\{x_t\}_{t=1}^T$ and one-step-ahead forecasting. Fix a history length $L \geq 1$. From the training sequence we construct triples of the form

$$\big(\mathbf{H}_j, y_j, \mathbf{F}_j\big) = \Big((x_j, x_{j+1}, \ldots, x_{j+L-1}),\ x_{j+L},\ (x_{j+L+1}, \ldots, x_{j+2L})\Big), \quad j = 1, \ldots, N.$$

Here $\mathbf{H}_j \in \mathbb{R}^L$ is the history window, $y_j \in \mathbb{R}$ is the one-step target, and $\mathbf{F}_j \in \mathbb{R}^L$ is a post-target continuation segment of the same length. This realises the natural "history–target–post-target continuation" decomposition at a minimal scale.

**Continuation summary.** For each training triple $(\mathbf{H}_j, y_j, \mathbf{F}_j)$ we define a scalar continuation summary

$$s_j = \frac{1}{L} \sum_{i=1}^{L} \big(F_j[i] - H_j[i]\big),$$

that is, the average increment between the aligned continuation and history entries. Intuitively, $s_j > 0$ indicates that, on average, the trajectory tends to keep moving upwards after the target, while $s_j < 0$ indicates a downward tendency, and $s_j \approx 0$ corresponds to approximately flat continuation. We then form a library of history–continuation summaries

$$\mathcal{D}_{\text{CS}} = \big\{(\mathbf{H}_j, s_j)\big\}_{j=1}^N.$$

**Linear continuation model.** As a simple generative model, we posit that the target $y_j$ can be expressed as a local adjustment of the last history value by the typical continuation tendency:

$$y_j = x_{j+L-1} + \beta\, s_j + \epsilon_j,$$

where $\beta \in \mathbb{R}$ is an unknown coefficient and $\epsilon_j$ is a zero-mean noise sequence with $\mathbb{E}[\epsilon_j \mid \mathbf{H}_j, s_j] = 0$. In this toy setting the continuation summary $s_j$ acts as a one-dimensional feature that modulates how the last history value should be adjusted. Given $(\mathbf{H}_j, y_j, s_j)_{j=1}^N$, the least-squares estimate of $\beta$ is

$$\hat{\beta} = \arg\min_{\beta \in \mathbb{R}} \sum_{j=1}^{N} \big(y_j - x_{j+L-1} - \beta s_j\big)^2,$$

which has a closed-form solution in terms of the sample covariance between $s_j$ and the residual $y_j - x_{j+L-1}$.

**Non-parametric estimation of continuation for new histories.** At test time, given a new history window $\mathbf{X} = (x_1, \ldots, x_L) \in \mathbb{R}^L$, we no longer have access to its true post-target continuation. However, we can estimate a typical continuation summary for histories similar to $\mathbf{X}$ by nonparametrically smoothing the values $\{s_j\}$ in the library $\mathcal{D}_{\text{CS}}$. Specifically, we use a Nadaraya–Watson kernel regressor (Nadaraya, 1964; Watson, 1964; Bierens, 1994):

$$w_j(\mathbf{X}) = \frac{K\big(\|\mathbf{X} - \mathbf{H}_j\|/h\big)}{\sum_{i=1}^N K\big(\|\mathbf{X} - \mathbf{H}_i\|/h\big)}, \qquad \hat{s}(\mathbf{X}) = \sum_{j=1}^N w_j(\mathbf{X})\, s_j,$$

where $K : \mathbb{R}_{\geq 0} \to \mathbb{R}_{\geq 0}$ is a bounded kernel (e.g. Gaussian) and $h > 0$ is a bandwidth. Here $\hat{s}(\mathbf{X})$ estimates the conditional mean $\mathbb{E}[s \mid \mathbf{H} = \mathbf{X}]$ based on the training library.

**Resulting predictor.** Combining the linear model with the non-parametric estimate of the continuation summary, we obtain the following simple continuation-based predictor:

$$\hat{y}(\mathbf{X}) = x_L + \hat{\beta}\, \hat{s}(\mathbf{X}),$$

where $x_L$ is the last entry of the history window $\mathbf{X}$. In words, $\hat{s}(\mathbf{X})$ captures how histories similar to $\mathbf{X}$ have typically continued beyond the target in the training data, and $\hat{\beta}$ determines how strongly this continuation tendency should influence the next-step prediction.

Under standard assumptions from non-parametric regression and linear models—such as Lipschitz continuity of the regression function $\mathbf{X} \mapsto \mathbb{E}[s \mid \mathbf{H} = \mathbf{X}]$, bounded support of the histories, a regular kernel $K$, and a bandwidth schedule with $h \to 0$ and $Nh^d \to \infty$—the Nadaraya–Watson estimator $\hat{s}(\mathbf{X})$ is a consistent estimator of $s^\star(\mathbf{X}) = \mathbb{E}[s \mid \mathbf{H} = \mathbf{X}]$, and the induced predictor

$\hat{y}(\mathbf{X})$ can be analysed within the same framework in terms of bias–variance trade-offs as $N \to \infty$ (Tsybakov, 2009; Györfi et al., 2002). A full statistical treatment of this distilled variant is beyond the scope of this paper, but we hope this toy model clarifies how continuation-style information can be incorporated into a mathematically coherent estimator "in the same vein" as KUP-BI, while abstracting away from the architectural details of modern deep backbones.

