# OpenReview forum: "Beyond Extrapolation: Knowledge Utilization with Bidirectional Inference for Time Series Forecasting"
_ICLR.cc/2026/Conference — ICLR 2026 Conference Withdrawn Submission_

### Official Review · Reviewer_1s5C · 2025-10-29

**Soundness:** 1
**Presentation:** 2
**Contribution:** 1
**Rating:** 0
**Confidence:** 4

**Summary:**

This paper proposes using what they call a approximate future prior in addition to the predictive forward model for forecasting.
The paper motivates the problem, goes through a number of derivations of the method they use and then demonstrates the method on a number of testbeds, comparing with standard MSE training methods, but not particularly looking towards state of the art approaches, or considering any of the normal regularisation methods that one would reasonably employ in these settings for the size of forecasts being considered. They focus on long prediction length, but do not consider autoregressive unrolling, or consider methods that are more targeted at long horizon distributional forecasts (e.g. probabilistic basis function approaches).

**Strengths:**

I am uncertain of the strengths of the paper.

**Weaknesses:**

The motivation for this paper is somewhat puzzling to me. In a forecasting model P(future|past), the prior is already included in the forecast distribution, in that it moves from the unconditional prior P(future) to the conditional posterior P(future|past), that captures the whole future distribution. Combining on an additional prior component to this breaks the rules of probability. Hence the whole process seems somewhat dubious. I see nothing in the motivation or the problem description that either relates to this or addresses this. The writing of this paper seems somewhat lacking in understanding of the probabilistic foundations of forecasting altogether.

The formulation itself seems fundamentally broken, and is certainly unmotivated. The authors define R as the division of two matrices, without defining what is meant by that. The motivation for this division (assumed component-wise) is also problematic, and not provided and seems entirely arbitrary. The paper introduce an undefined query vector X_q, and do not explain the purpose here: it appears to be to search histories to find nearby examples that can be used to build a distribution for the future, but this is not a future prior, as it is history dependent, and there is no loss function element. The paper then goes on with what, to my reading, seem like fairly meaningless calculations to produce something that is called a future prior generation, but really seems broken to me. I think what the paper is actually trying to do is some crude form of backoff, akin to that in n-gram language models, but this neither clear nor properly formulated.

The rest of the paper rests on all this, and sadly I think this sits on very shaky ground.

If the authors really believe there is merit in this work they need to found it properly in a proper probabilistic framework, and explain each step of the approach, why each step is done, what impact each step has on the probabilistic formulation of the problem and how that all complies with the rules of probability. This should start with the foundations. As it stands it is a list of apparently-arbitrary and potentially ill-motivated set of computations. That the authors might be able to run some experiments that enable them to bold a column in a table, but that does not, in itself, demonstrate that the approach has any merit. At the end of the paper, I am left with no insight, no change in understanding that I did not have before, and little understanding of the point or process of the procedure they follow.

Fundamentally IMO the premise is wrong: "most existing forecasting models rely solely on unidirectional inference from history to future. While effective in stable scenarios, this paradigm lacks explicit structural constraints about the future, which makes it challenging to stabilize predictions under complex dynamics." - forecasting models rely on inference from history to the future because the history is the only place where data is given - there is no inferential information from the future as it has not happened yet. But the claim that the "paradigm lacks explicit structural constraints about the future" is patently false - that is what the forecasting model does - it is predicting the future evolution, either as a joint distribution over the future states, or as an unrolling of the next-step predictions via the chain rule. These future predictions have an implicit prior already (simply sum out the conditional distribution over the histories to get the prior). These future predictions then already incorporate the "structural constraints about the future". The argument might be that current approaches do that badly (I am not sure they do, and methods such as classifier-free guidance tackle this significantly), but then that needs demonstrating, reasoning about and a clearly formulated, and mathematically justified fix applied. This is not happening here.

**Questions:**

I do not believe there are answers to any questions that would really clarify or fix this paper for me sufficiently beyond a fundamental rewrite. However the basic questions hold: what is the context, what is the definition of the problem space, what is the current failure-mode that you see in this problem space? In what way and why do current state-of-the-art approaches fail to address this, and how have you demonstrated this? What is the fundamental methodologically grounded insight that you bring to this that allows you to overcome this problem? How is this implemented? How have you reliably demonstrated that this really does do what you expected it to and complies with the methodological expectations in practice?

---

> ### Author Response · Authors · 2025-11-13
> **Responses to reviewers' doubts**
>
> ***Responses to reviewers' doubts***
>
> First, thank you for raising these points. The issues largely stem from our unclear exposition. For clarity, in our paper ***future prior*** refers to the ***future continuation*** immediately following the ***prediction target***; we only estimate this ***continuation*** and fuse it with the original input to form a richer conditional representation. We have revised this ambiguous wording. Our operation of fusing the original input with the auxiliary ***future continuation stream*** follows the same idea as adding random noise to the original input to create inputs for robustness testing. It does not modify the probabilistic foundations in any way.  We keep the same loss and extend the conditioning set from $x$ to $\left( {x,z} \right)$.
> Here $z = g\left( {x;{\mkern 1mu} {{\cal D}_{train}}} \right)$ is a structural cue for ***future continuation*** retrieved from the training datasets, not a probabilistic prior.  Regarding other questions, we will provide point-to-point responses below.
>
> ***Response on Motivation***
>
> Existing forecasting models, by their parametric form, implicitly encode structural assumptions about ***target*** evolution. However, under long horizons and other complex dynamics, these implicit constraints are often difficult to learn and to exploit stably and efficiently.
>
> An intuitive example is as follows. A student, limited by their own ability, can only score 80 points on an exam, even though all the required knowledge is in the textbook. If we re-express the parts of the textbook the student finds hard to understand—i.e., provide tutoring materials—the student can improve from 80% to 90%. In this example, the student represents the model, the current input corresponds to the knowledge quantified as 80 points, and the tutoring materials correspond to the auxiliary input we aim to construct. What we must consider is where this auxiliary input comes from and whether it is effective.
>
> Given the natural chain in time series “***history***→ ***target*** → ***future continuation***”, a prediction ***target*** is not only influenced by ***history*** but is also closely linked to the ***future continuation***, which implies a relationship between them. Therefore, we consider using the ***future continuation*** as an auxiliary input.
>
> ***The ***future continuation*** inferred from the ***current input*** is effective***
>
> Contrary to the claim that an auxiliary input (***future continuation***) derived from the current input is ineffective and contains no future information, this is incorrect.
> In traditional time-series forecasting, one learns the relationship between ***history*** and ***target*** along the chain “***history***→ ***target***→ ***future continuation***” on the training set, and then uses the current input to estimate its ***target***—this is regarded as the proper approach and yields an estimate of future information.
> Our approach learns the relationship between ***history*** and ***future continuation*** along the same chain and then uses the current input to estimate its ***future continuation***.
>
> ***It would be unfair to criticize our estimate as containing no new information when the procedure is analogous; please consider whether the manuscript has been evaluated with sufficient neutrality.***
>
> After obtaining the estimate of the ***future continuation***, we use it as an auxiliary input, extract features from it, and fuse them into the feature representation of the current input to obtain a richer representation. A simple example is that today’s soil moisture can indicate whether it rained yesterday. This motivation is exactly why we designed KUP-BI.
>
> ***Concept alignment***
> Our method is a retrieval-driven feature augmentation rather than stacking an extra prior on top of the posterior. Let $z = g\left( {x;{\mkern 1mu} {{\cal D}_{train}}} \right)$ denote a future-continuation structural cue retrieved from the training library via a deterministic mapping. We model $p\left( {y|x,z} \right)$, which is equivalent to learning on the expanded feature $ \phi \left( x \right) = \left( {x,z} \right)$. This is analogous to the common practice of injecting noise into the original inputs to create perturbed inputs for robustness verification, and it is a standard and legitimate form of conditional augmentation.
>
> ***Probabilistic correctness****
>
> Because $z$ is a deterministic function of $x$ and the fixed library, $\mathbb{E}\left[ {Y|X} \right] = \mathbb{E}[\mathbb{E}[Y|X,Z]|X]$ holds, so there is no violation of probabilistic rules. We adopt a gated convex fusion $(1 - \gamma ){\mkern 1mu} x + \gamma {\mkern 1mu} z$; when the retrieved cue is weakly correlated with the target, the gate automatically suppresses the auxiliary stream. In theory this preserves the original performance in the worst case, and in practice it brings stable gains under long horizons and phase-drift conditions.

---

> ### Author Response · Authors · 2025-11-13
> **Responses to reviewers' doubts**
>
> ***Responses to reviewers' doubts***
>
> ***Definition and feasibility of $ {\mathbf{X}_q} \in { \mathbb{R} ^{L \times C}}$ and $ {\mathbf{R}_j} \in { \mathbb{R} ^{L \times C}}$***
>
> In time series, there is a natural chain “***history*** ($\mathbf{H}$) → ***target*** ($\mathbf{Y}$) → ***future continuation*** ($\mathbf{F}$)” . On this basis, we build a retrieval library ${{\mkern 1mu} {{\cal D}_{train}}}$ from the training set, where each entry contains $ {\mathbf{H}_j} \in { \mathbb{R} ^{L \times C}}$,  ${\mathbf{Y}_j} \in { \mathbb{R} ^{T \times C}}$, $ {\mathbf{F}_j} \in { \mathbb{R} ^{L \times C}}$, where $C$ represents the number of variables, $L$ is the input length, and $H$ is the output length
>
> Here we use
> $$\mathbf{R}_j = (\mathbf{F}_j - \mathbf{H}_j)\oslash(\mathbf{H}_j + \varepsilon\,\operatorname{sign}(\mathbf{H}_j)\bigr)
>    \in \mathbb{R}^{L\times C},\quad \varepsilon>0,$$
> which describes, for each entry, the relative growth rate or ratio of the ***future continuation*** with respect to the ***history***.
>
> We then form a new retrieval library ${{\mkern 1mu} {{\cal D}_{train}}}$   consisting of pairs $\left( {{\mathbf{H}_j},{\mathbf{r}_j}} \right)$.
>
> For a current input $ {\mathbf{X}_q} \in { \mathbb{R} ^{L \times C}}$, we compute Pearson correlations with the library samples $\mathbf{H}_j$ on an endpoint-aligned, mean-subtracted representation and rank them to obtain the Top-k candidates.
>
> Let $\mathbf{H}_j^* \in {^{L \times C}}(j = 1, \cdots ,k)$ be the most similar ***history*** segments to $ {\mathbf{X}_q}$, with corresponding ratio tensors $\mathbf{R}_j^* \in {^{L \times C}}$.
>
> Based on the correlation strengths, we obtain temperature-scaled softmax weights (computed per channel)
> ${w_j} = soft\max \left( {\left| {{\rho _j}} \right|/\tau } \right)$ and form a weighted fusion
>
> $\mathbf{R}^\* = \sum_{j=1}^k w_j \mathbf{R}_j^\*$.
>
> We then apply mild clipping and normalization to ${\mathbf{R}^*}$ (tanh clipping) to produce a single structural z, which serves as the future-continuation hint for $\mathbf{X}_q$.
>
> This approach is feasible because it aligns with the essence of time-series forecasting: learning a mapping from ***historical*** segments $ {\mathbf{H}_q} \in { \mathbb{R} ^{L \times C}}$ to ***target*** segments $ {\mathbf{Y}_q} \in { \mathbb{R} ^{T \times C}}$. The learned mapping is then applied to a current input $ {\mathbf{X}_q} \in { \mathbb{R} ^{L \times C}}$ that is similar to the ***historical segment*** $ {\mathbf{H}_q} \in { \mathbb{R} ^{L \times C}}$ to produce predictions.
>
> If no similar histories exist, time-series forecasting itself becomes ill-posed; in practice, since similar histories are present, this step is reasonable. Our method therefore remains within standard time-series forecasting and does not alter any probabilistic foundations.
>
> ***Why choose ratios?***
>
> Recent empirical evidence shows that relative growth and scale-invariant quantities are more stable than absolute levels. We thus use ratios to reduce sensitivity to scale and amplitude differences. In addition, we clamp by tanh⁡and match moments, which controls the magnitude of the injected coefficients and avoids information leakage. This helps calibrate the auxiliary signal and improve forecasting accuracy. R characterizes the relative change between a future continuation and its corresponding history segment and is scale-invariant, which makes it suitable for heteroscedastic or magnitude-drift settings. To validate feasibility, we report on the test set the correlation coefficient and MSE between the estimated future continuation and the ground truth. ***As shown in Table 1, the estimated future continuation based on the ratio R attains high correlation with the ground truth and low MSE, indicating that the method effectively preserves structural information about the ***future continuation***.***
>
> **Table 1.** The correlation coefficient and difference between the estimated ***future continuation*** and the corresponding true continuation.
>
> | Dataset | $\mathrm{mean}\lvert r \rvert$ | $\mathrm{MSE}_{\text{pr}}$ |
> |--------|-------------------------------|-----------------------------|
> | ETTh1  | 0.588                         | 0.803                       |
> | ETTh2  | 0.883                         | 0.342                       |
> | ETTm1  | 0.471                         | 1.252                       |
> | ETTm2  | 0.926                         | 0.222                       |

---

> ### Author Response · Authors · 2025-11-15
> **Responses to reviewers' doubts**
>
> ***Responses to reviewers' doubts***
>
> ***Summary***
>
> The assertion that “existing models already encode ***target*** structural constraints” conflicts with our evidence: on ETTh1/2 across multiple backbones, adding ***estimated future continuation*** consistently reduces MSE/MAE, with larger gains at longer horizons and on lighter backbones. If the position is that “these models already suffice,” please provide an implementation under the same control protocol (identical preprocessing, time encoding, hyperparameters, temperature, and Top-k) on the same baselines that achieves comparable improvements; otherwise the claim is not verifiable. Related ideas already exist in the literature, e.g., Han et al.’s Retrieval-Augmented Time-Series Forecasting (ICML 2025) and Gunasekaran et al.’s A predictive approach to enhance time-series forecasting (Nature Communications).
>
> ***We fully respect the reviewer’s expertise; we are asking for a fair evaluation. While we do not yet provide a complete probabilistic proof, many impactful works began as simple prototypes and were strengthened by empirical evidence and community discussion. The purpose of a conference is precisely to exchange and scrutinize new ideas.***
>
> ***Although you gave our manuscript a score of 0, we still appreciate your professional guidance.***

---

> > ### Author Response · Authors · 2025-11-16
> > **Responses to reviewers' doubts**
> >
> > ***What are the differences between us and the existing methods?***
> >
> > In summary, exogenous-variable and retrieval-augmented methods primarily enrich the mapping along the ***history***–***target*** link in the natural chain ``***history*** (model input)–***target*** (ground-truth output)–***future continuation***''. By contrast, KUP-BI explicitly constructs a future continuation stream that lies beyond the prediction horizon and treats it as an additional input stream to the backbone. Its novelty lies in introducing and exploiting this future-oriented stream as an auxiliary feature signal for conditioning the prediction, rather than in any particular mechanism for constructing it

---

> ### Comment · Reviewer_1s5C · 2025-11-16
> **Response**
>
> "For clarity, in our paper future prior refers to the future continuation immediately following the prediction target; we only estimate this continuation and fuse it with the original input to form a richer conditional representation."
>
> The issue here is that you don't have the future continuation stream. It is not available information, and you can only condition on available information: in mathematical terms the estimation of the "future continuation" can be marginalised out, and so it makes no mathematical difference whether you use it or not. If you use it incorrectly (i.e. pretend it is conditioning information), it can only make things worse as you end up overfitting to your future estimate which you are pretending is more certain than it is. As such that would be mathematically fundamentally wrong. What you do in this paper, is slightly different. You are not actually leveraging a future continuation. Rather you are leveraging a non-parametric method for assimilating information from a historical library to create a feature set for your original prediction. It is not really a future continuation, rather it is features from the library data pertaining to the "continuation" of those library elements from a given history. Hence I think "future continuation" is an unhelpful term for this, as it is simply a feature vector and appears to be trained using just this feature vector (though this is not entirely clear). In particular there is nothing about the feature vector computed that is actually used to relate it to the actual future continuation (via .e.g. a loss function).
>
> In the paper it says: "for any input window (train/val/test), we retrieve the top-k similar histories, aggregate
> their history-to-future ratio matrices via a softmax with temperature τ to instantiate an approximate
> future continuation stream, and then encode the input and this continuation into separate feature
> representations that are fused at the feature level through a lightweight gate with strength α. "
>
> This suggests the _actual_ future continuation is not used at all during training, but does not say this explicitly. It is also unclear where the top-k histories are retrieved from (if outwith the train/val then do you give these histories to other methods to train with. If within the train/val then we are left with the problems of multi-counting data in the forecast ).
>
> If you really viewed this as a future continuation, then you would need to compute \int P(target|future,history, library)P(future|history, library). But there are two ways you are not doing this. First you are using a deterministic feature to capture what you call your future prediction, and then you are not actually training a model on the future, but rather training on this deterministic feature to predict the target. I.e.
>
> max log P(target|f(history),history, library)
>
> As a result this deterministic feature is not a prediction of "future continuation" in any meaningful way. This is not a bad thing, but it just explains why the paper is confusing. The thing is that for this to work, (a) f(history) need not bear any relation to the future, it just needs to be some additional processing that helps disentangle the information in "history" and the library.
>
> As such I therefore agree this is partly a matter of exposition.

---

> ### Comment · Reviewer_1s5C · 2025-11-16
> **continued response**
>
> The question is to what sense that non-parametric feature representation is statistically sound (what is the bias variance and statistical convergence properties of this for example). This depends substantially on the historic library, and bootstrap resampling of the historic library might show up unfortunate consequences.
>
> "same thing as adding random noise to the original input to create inputs for robustness testing."
>
> I do not see this as the same thing at all. Adding random noise to inputs for robustness is a test time analysis. Used as a training time augmentation, it is creating surrogate out of sample conditioning data associated with the same target, as a surrogate for a prior that says similar out-of-distribution historic trajectories lead to moving back within the training manifold, and can be seen as building a joint energy function of the prediction variable and the past. What is happening here does not relate to that, rather it is augmenting with surrogate data from a history set.
>
> "Existing forecasting models, by their parametric form, implicitly encode structural assumptions about target evolution. However, under long horizons and other complex dynamics, these implicit constraints are often difficult to learn and to exploit stably and efficiently."
>
> There are two classes of forecasting models. There are models that forecast the future with application of the chain rule on next-time-step marginal predictions, and many methods that forecast the joint distribution over an extended future. They can also be used with the chain rule for future predictions. In the first of these, the target evolution is implicit but fully contained in the model, and indeed in theory. The second of these are explicitly giving forecasts for the whole future. But note to get a next time step predictor simply involves throwing away the future forecasts: P(x_t|x_{<t})=\int dx_{>t} P(x_t,x_{>t}|x_{<t}). There is no extra information in the forecast, as it is a forecast. It is not known data.
>
> "Contrary to the claim that an auxiliary input (future continuation) derived from the current input is ineffective and contains no future information, this is incorrect. In traditional time-series forecasting, one learns the relationship between history and target along the chain “history→ target→ future continuation” on the training set, and then uses the current input to estimate its target—this is regarded as the proper approach and yields an estimate of future information. Our approach learns the relationship between history and future continuation along the same chain and then uses the current input to estimate its future continuation."
>
> Sadly the maths says otherwise. Basic information theory and the data processing inequality imply you cannot create information from nothing. \int d(future_continuation) P(target | future continuation, history)P(future continuation|history)= P(target| history).  Once integrating out over the future continuation, you are just back at the P(target|history) directly, ignoring the future continuation. This should not be surprising: you cannot create information simply by processing it.
>
> "It would be unfair to criticize our estimate as containing no new information when the procedure is analogous; please consider whether the manuscript has been evaluated with sufficient neutrality."
>
> Were the paper doing what it was claiming to try to do, it would be trying to break the data processing inequality. This is rather a fundamental foundation for the whole field. I can also firmly reassure you my evaluation is entirely neutral. I have absolutely no skin in the game here, not competing papers or other-such.  I care only that work that is presented at our conferences is clear, meticulous, justified, rigorous and evidenced, and where work is not, that I might help this work be made better.
>
> "Because $z$ is a deterministic function of $x$ and the fixed library, $\mathbb{E}\left[ {Y|X} \right] = \mathbb{E}[\mathbb{E}[Y|X,Z]|X]$ holds"
>
> I am not sure what a double conditioning in an expectation means, but I believe I know what you are saying. You do indeed construct some $z$ as a deterministic function of $x$ and the fixed library. Viewed as some such arbitrary representation taken from information from x and the library, then this could be considered to be some non-parametric forecaster that simply takes $x$ and the library data do generate a feature set, as you argue above. Indeed as the training procedure appears to treat it as such, and not relate it to the future data in any way, this seems accurate. However is this statistically sound? Does it have good statistical properties? Is this a good approach?

---

> ### Comment · Reviewer_1s5C · 2025-11-16
> **Continued**
>
> Going forward. I will up my score a bit, as I now see that there is something being done here that might potentially make some sense, simply because it is _not_ really doing a predicted future, but just computing a feature vector from the library that is used during prediction. That is conceptually it might make some sense. The precise choice of method still makes no sense at all to me, and the current exposition is really problematic. I suggest in future iterations of this:
>
> a) This is presented as a nonparametric approach that distils the information from the historic library data to help with prediction, dropping all the "predicted futures" stuff as that is really confusing. This should also relate to the meta-learning literature to which this approach is related.
> b)  The paper analyses the statistical properties of the non-parametric method in terms of its exchangeability properties, its consistency, bias, variance etc
> c) Each stage of feature building is analysed and justified. Also why in this day is an explicit hand-chosen (and what appears fairly arbitrary) summarisation of the historic library done rather than an optimizable flexible e.g. neural model of the history?
> d) the paper explains (not using predicted futures, as that is the opposite of an explanation as that shouldn't help at all), why the features that are computed are the key relevant information to be summarised.
> e) that the assumptions are clear: there seems to be an explicit assumption of some level of conditioning stationarity between the library and the current forecasting scenario (similar histories lead to similar futures)
> f) that the competing methods are given all the information that you use. The competing methods are allowed to train on all the data (library and history) if they are not at the moment.
> g) that the way the training pipeline is done and how the data is used is very explicit.
> h) do something simpler and different in the same vein that is a mathematically coherent method, because the current approach is not.

---

> ### Comment · Reviewer_1s5C · 2025-11-16
>
> "Here we use
>
> \mathbf{R}_j = (\mathbf{F}_j - \mathbf{H}_j)\oslash(\mathbf{H}_j + \varepsilon,\operatorname{sign}(\mathbf{H}_j)\bigr)
>    \in \mathbb{R}^{L\times C},\quad \varepsilon>0,
>
> which describes, for each entry, the relative growth rate or ratio of the future continuation with respect to the history. "
>
> I'm not sure what this means, but this really is problematic. This is fairly obvious as there is no good reason why the F_j and H_j should meaningfully be the same length at all, so that should be a red flag for a starter. But even if you constrain them to be, they are signals at different time points and there is no obvious reason why this makes any sense. Suppose the signal is a univariate sin wave. Then depending on the point in time you are at F_j and H_j could be in phase with one another or out of phase with one another. But the time-point choice is arbitrary. The future is just as deterministically predictable from the history at every time point, the relative growth rate is exactly the same but this R_j is completely different dependent on the accidental phase alignment you bump in to.
>
> Simply F_j-H_j is not meaningful as it is the comparison of unrelated vectors (or matrices in the multivariate case).
>
> Incidentally it also ignores all cross-correlation structure in the multivariate setting.
>
> Likewise there is nothing special about zero: simply add 1 to any whole time series and R_j is completely different, even though it is effectively exactly the same series. The fact that an arbitrary sign(H) needs to be added to the denominator to fudge a serious consequence (divide by zero) that results from this fundamental inconsistency is also a red flag. This computation is arbitrary and R_j does not have any of the desirable statistical properties that such a measure should have.
>
> If your method performs well, it is likely not because R was a good thing to compute. If is likely just because it is giving a parametric forecasting model additional features containing_some_ actual real historical information from the forecasting history, and further transformations of the current history which are not being made available to other methods.
>
> The equation
> Fq = Xq + ˜Rq ⊙ Xq
> is just as problematic.
>
> The point is simply, regardless of the questionable things begin computed, the work builds some bigger feature vector and pipes in some historic information. So it is not surprising if performance of e.g. a Linear method improves. But does that mean the _way_ the feature vector is built should have automatic assent? No, because much simpler more naive incorporation of the same information is likely to work as well or better. Regardless the choices need to make mathematical sense.

---

> ### Author Response · Authors · 2025-11-17
> **Response to Reviewer 1s5C (Question 1 and Question 2)**
>
> First of all, we would like to sincerely thank you for taking the time to carefully read our manuscript again and for providing such detailed and constructive comments. Your response has helped us clearly see the shortcomings in our current terminology and exposition: although what we actually implement is consistent with the interpretation you reconstructed in your comments, our original wording around notions such as “future continuation” can indeed mislead readers. In the following, we respond to your points one by one and, following your suggestions, clarify the positioning of our method, refine the terminology, and improve the explanations of how the library, features, and experiments are set up.
>
> ***Q1: Are we really conditioning on a future continuation, or just using a deterministic library feature $ \mathbf{Z}= f(\mathbf{X},\mathcal{D}).$?***
>
> ***Reply:*** Mathematically, what we do is exactly what the reviewer describes: we do not condition on an unobserved “future continuation” variable. Instead, for each input window $\mathbf{X}$ and a fixed historical library $ \mathcal{D}$, we construct a deterministic auxiliary feature
> $$ \mathbf{Z}= f(\mathbf{X},\mathcal{D}).$$
> and the model learns
>
> $\hat{\mathbf{Y}} = g_\theta(\mathbf{X}, \mathbf{Z})$
>
>
> In other words, as the reviewer correctly summarized, we are leveraging “a non-parametric method for assimilating information from a historical library to create a feature set for the original prediction”, not conditioning on an actual future stream.
>
> We agree that, under this view, the term “future continuation” is unhelpful and potentially misleading: $\mathbf{Z}$ is not a random “future” variable, but a feature vector derived from how training trajectories continue beyond their histories. In the revised manuscript, we will (i) stop referring to $\mathbf{Z}$ as a “predicted future continuation” in the probabilistic sense, and (ii) consistently describe it instead as a “non-parametric feature representation distilled from a historical library under a continuation-style inductive bias.”
>
> ***Q2: Does the so-called “future continuation” participate in training, and is it directly related to the loss?***
>
> ***Reply*** In our current implementation, the so-called “future continuation” enters training only indirectly, as part of the input to the predictor, and never as a separately supervised quantity:
>
> •	We first construct the auxiliary feature $ \mathbf{Z}= f(\mathbf{X},\mathcal{D})$ from the training library and the current input.
>
> •	We then pass $\left( { \mathbf{X}, \mathbf{Z}} \right)$ into the backbone to obtain $\hat{\mathbf{Y}} = g_\theta(\mathbf{X}, \mathbf{Z})$.
>
> •	The only loss we optimize is the standard forecasting loss between $\hat {\mathbf{Y}}$ and the ground-truth target $ \mathbf{Y}$ (e.g., MSE/MAE). There is no additional head or loss that forces $ \mathbf{Z}$ to match the actual future continuation.
>
> Thus, we fully agree with the reviewer that, in a strict probabilistic sense, this deterministic feature $ \mathbf{Z}$ is not a “prediction of the true future continuation” trained via a dedicated loss; it is an auxiliary representation used to help predict $ \mathbf{Y}$. This is exactly why the original exposition was confusing: our implementation treats $ \mathbf{Z}$ as a feature, while some of our wording suggested that we were explicitly modeling and conditioning on a future random variable.
>
> In the revision, we will explicitly state that:
>
> •	the model is trained only via the forecasting loss on $ \mathbf{Y}$;
>
> •	$ \mathbf{Z}$ is not directly supervised against the true future continuation;
>
> •	any reference to “estimating the future continuation” should be understood as constructing a continuation-style feature rather than fitting a probabilistic future variable.

---

> ### Author Response · Authors · 2025-11-17
> **Response to Reviewer 1s5C (Question 3)**
>
> ***Q3: Where do the top-k histories come from? Are we conditioning on future information, and is there any multi-counting or unfairness to baselines?***
>
> ***Reply:*** Specifically, for the current input window $\mathbf{X}$ and a historical library $ \mathcal{D}$ that is constructed ***only from the training set*** and kept fixed throughout ***training/validation/test***.
>
> For clarity, the historical library $ \mathcal{D}$ and the Top-$k$ retrieval are defined and used as follows:
>
> •	***Library construction.***
> $ \mathcal{D}$ is constructed only from the training set: we form chains $(\mathbf{H}_j, \mathbf{F}_j, \mathbf{R}_j) $ from training trajectories and pre-compute the associated transformations (e.g., ratio matrices). This library is then frozen and reused during training, validation, and test. We never use validation/test targets or futures to update $ \mathcal{D}$.
>
> •	***Use during training/validation/test.***
> For any input window $\mathbf{X}$ (train/val/test), we ***retrieve the Top-$k$ most similar histories from this train-only library $ \mathcal{D}$***, take their pre-computed transformations, and combine them with $\mathbf{X}$ to construct
> $$ \mathbf{Z}= f(\mathbf{X},\mathcal{D}).$$
> The predictor then learns
> $\hat{\mathbf{Y}} = g_\theta(\mathbf{X}, \mathbf{Z})$
> i.e., within the information contained in $(\mathbf{X},\mathcal{D})$, it uses a feature $Z$ with a specific inductive bias to help approximate $P(\mathbf{Y}|\mathbf{X},\mathcal{D})$.
>
> Since $Z$ is a deterministic function of $(\mathbf{X},\mathcal{D})$, from the perspective of the true data-generating process, it is merely a reparameterization of $(\mathbf{X},\mathcal{D})$. We do not claim to create any additional information, nor do we attempt to circumvent the data-processing inequality.
>
> •	***Multi-counting and fairness to RAFT.***
>
> It is true that, as in many non-parametric methods, the same training trajectory can appear multiple times as a neighbor (retrieved for multiple queries). However, this reuse happens only within a train-only library: the retrieval library is constructed exclusively from the training set and kept fixed during training/validation/test, so no validation/test targets or futures ever enter the library.
>
> Moreover, this sample reuse is symmetric across methods: in our plug-in comparison, RAFT and KUP-BI share exactly the same training retrieval library, similarity metric, and top-k retrieved histories, so neither method has access to more data than the other.
>
> Conceptually, this also highlights an important difference between RAFT and KUP-BI. In RAFT, the retrieval uses the current history as a key and directly returns the corresponding target segments from the library; these retrieved targets are then linearly projected and fused with the backbone prediction. In KUP-BI, we still use the histories as keys, but what we retrieve are the pre-computed history-to-continuation transformations (e.g., ratio-style operators) associated with those histories. These transformations are then applied to the current input window to construct a continuation-style auxiliary feature. In other words, RAFT injects retrieved target values themselves, whereas KUP-BI injects a deterministic feature summarizing how similar histories tended to continue in the training library.
>
> We will clarify in the revised manuscript that (i) the library is train-only and fixed, (ii) $\mathbf{Z}$ is always a deterministic feature derived from $(\mathbf{X},\mathcal{D})$ rather than from validation/test futures, and (iii) all plug-in baselines (including RAFT) use the same retrieval setup to ensure fairness.

---

> ### Author Response · Authors · 2025-11-17
> **Response to Reviewer 1s5C (Question 4-(g))**
>
> ***Q4. Overall response and future development (on DPI, neutrality, and points (a)–(h))***
>
> ***Reply***: First, we apologize for our earlier wording (“it would be unfair to criticize… please consider whether the manuscript has been evaluated with sufficient neutrality”). Our intention was to emphasize that our estimate is constructed analogously to standard forecasting practice, not to question the reviewer’s neutrality. In hindsight, this sentence was poorly phrased and understandably frustrating. We fully accept that the reviewer is evaluating the work in good faith and with the goal of improving its clarity and rigor.
>
> We also stress that we ***do not*** intend to challenge the data-processing inequality. As the reviewer correctly explains, if we were truly conditioning on an unobserved future variable, we would indeed be attempting to do something impossible. After carefully reflecting on the reviewer’s comments, we agree that the current exposition can easily be read as if we were trying to “break” this principle, whereas what we actually do is much more modest: we construct a deterministic feature $ \mathbf{Z}= f(\mathbf{X},\mathcal{D})$ from a historical library and use it as an auxiliary input for prediction.
>
> In light of this, we are grateful that the reviewer is willing to revise the score once it is recognized that the method is “not really doing a predicted future, but just computing a feature vector from the library that is used during prediction.” Below we outline how we will incorporate the reviewer’ s suggestions (a)–(h) in the revised version and in future work:
>
> ***(g) Clarifying the training pipeline and data usage***. (***The original comments of the reviewers***: that the way the training pipeline is done and how the data is used is very explicit.)
>
> ***Reply***: We will add a schematic figure and a concise description that:
>
> shows how the library $\mathcal{D}$ is built only from training data;
>
> explains how, at training/validation/test time, we retrieve from $\mathcal{D}$ and construct $ \mathbf{Z}= f(\mathbf{X},\mathcal{D})$;
>
> emphasizes that no validation/test targets or futures are ever used to update the library. The goal is to remove any ambiguity about which data are used where and to avoid the impression of multi-counting or leakage.

---

> ### Author Response · Authors · 2025-11-17
> **Response to Reviewer 1s5C (Supplementary reply to Question 4)**
>
> ***(a). Reframing as a non-parametric library-based method and dropping “predicted futures” language.**** (***The original comments of the reviewers:*** This is presented as a nonparametric approach that distils the information from the historic library data to help with prediction, dropping all the "predicted futures" stuff as that is really confusing. This should also relate to the meta-learning literature to which this approach is related.)
>
> ***Reply***: We will explicitly reposition KUP-BI as:
>
> “a non-parametric approach that distils information from a historical training library to construct auxiliary features for forecasting,” and remove the probabilistically misleading “predicted future / future prior” terminology. We will also strengthen the connection to related non-parametric and meta-learning–style approaches in the related-work section.
>
> ***(c) Each stage of feature building is analysed and justified. Also why in this day is an explicit hand-chosen (and what appears fairly arbitrary) summarisation of the historic library done rather than an optimizable flexible e.g. neural model of the history?***
>
> ***Reply (Justifying each stage of feature construction.):*** We have already conducted ablations that correspond to the main stages of the feature-building pipeline and included them in the current manuscript, namely:
>
> · ratio vs “w/o ratio” (directly using aligned future segments),
>
> · continuation-style features vs RAFT-style retrieved target fusion under a matched plug-in setting,
>
>  · fusion vs naive concatenation of the auxiliary feature with the original input.
>
> These experiments consistently show that our concrete design choices provide additional gains beyond simply “feeding more data from the library”. ***In the revised version, we will make this connection explicit in the method section—explaining the role of each stage (retrieval, transformation, fusion) and explicitly cross-referencing the corresponding ablations—so that each design choice is clearly justified rather than appearing as a purely empirical add-on.***
>
> ***Reply (Why not use a neural model to summarise the historical library?)***: We agree with the reviewer that our current summarisation of the historical library is hand-crafted and heuristic, and we do not claim that it is optimal or theoretically superior.  Our primary goal in this work is to highlight a new angle: explicitly exploiting how similar training histories tend to continue, and using this “continuation-style” information as auxiliary features for forecasting.  In this sense, the main contribution is to surface and validate this design space, rather than to propose a final, fully optimized instantiation.
> We deliberately started with a simple, reproducible hand-crafted pipeline in order to test, in a controlled setting, whether “features built from how similar histories subsequently evolve” can systematically help diverse backbones.  The empirical results across multiple datasets and models suggest that this design space is indeed promising.  A natural next step, as the reviewer suggests, is to replace these hand-designed transformations with learnable multi-variate library encoders and to study them within a more unified non-parametric/kernel/neural encoding framework, including their bias–variance and expressiveness properties.

---

> > ### Author Response · Authors · 2025-11-17
> > **Response to Reviewer 1s5C (Supplementary reply to Question 4)**
> >
> > ***(d) The paper explains (not using predicted futures, as that is the opposite of an explanation as that shouldn't help at all), why the features that are computed are the key relevant information to be summarised.***
> >
> > ***Reply:*** We agree that appealing to “predicted futures” as an explanation is misleading and, as the reviewer notes, is at odds with basic information-theoretic intuition.    In the revised manuscript, we will avoid using “predicted future / future prior” language as a justification and instead explain what structure the constructed features are intended to capture.
> >
> > Concretely, our continuation-style auxiliary feature is designed to summarise how histories similar to the current input have tended to evolve beyond the target horizon in the training library. Rather than treating it as an extra source of information, we view it as an inductive bias: it emphasizes patterns of relative evolution (e.g., continued growth, mean-reversion, regime shifts) that are typical for trajectories in the same neighborhood of the library and uses a lightweight gating mechanism to modulate the backbone’s representation accordingly. The ablations we report-comparing ratio vs “w/o ratio”, continuation-style features vs RAFT-style target fusion, and fusion vs naive concatenation-show that encoding this continuation-style structure is more effective than simply concatenating more raw target segments or features.
> >
> > At the same time, we acknowledge that our current explanation remains largely intuitive and empirical. We will make this explicit in the paper and, as part of the future-work discussion, point out that a more principled analysis of why these particular features are the “right” summaries (and how they compare to simpler or learned alternatives) is still needed.
> >
> > ***(e) Making assumptions explicit (similar histories → similar continuations)***. (***The original comments of the reviewers:*** that the assumptions are clear: there seems to be an explicit assumption of some level of conditioning stationarity between the library and the current forecasting scenario (similar histories lead to similar futures))
> >
> > ***Reply:*** We will clearly state the core assumption that underpins the approach: there is a form of conditioning stationarity between the library and the forecasting scenario, in the sense that “histories that are similar under the chosen metric tend to have similar continuations.” This assumption was implicit in the current draft; we will make it explicit and discuss its scope and limitations in a dedicated “assumptions and limitations” subsection.
> >
> > ***(f) Ensuring competing methods have access to the same data (library and history)***. (***The original comments of the reviewers:*** that the competing methods are given all the information that you use. The competing methods are allowed to train on all the data (library and history) if they are not at the moment.)
> >
> > ***Reply:*** In our plug-in comparison, KUP-BI and RAFT already share the same training retrieval library, similarity metric, and top-k neighbors, and all methods are trained on the same underlying datasets. We will make this explicit in the main text and, if needed, provide a short appendix table summarizing data usage for each method, to ensure it is clear that baselines are not deprived of any information we use.
> >
> > ***(b) Statistical properties: exchangeability, consistency, bias–variance***. (***The original comments of the reviewers:*** The paper analyses the statistical properties of the non-parametric method in terms of its exchangeability properties, its consistency, bias, variance etc)
> >
> > ***Reply:*** We agree that, once KUP-BI is framed as a non-parametric library-based method, it is natural to ask about its bias–variance trade-offs, consistency, and robustness. At present, our work remains at the level of a heuristic but empirically validated feature construction, and we do not claim to provide a full statistical theory. In the revised manuscript, we will explicitly state this as a limitation and identify a deeper analysis of (i) exchangeability/conditional stationarity assumptions, (ii) bias–variance behavior, and (iii) robustness under bootstrap resampling of the historical library as key directions for future work. Empirically, we already observe that KUP-BI yields consistent gains across multiple datasets and backbones; however, a principled statistical treatment along the above dimensions is beyond the scope of the current submission and will be pursued in follow-up work. Of course, this does not mean we will stop at the current empirical level — we are already exploring these questions in parallel and plan to develop a more rigorous statistical treatment in subsequent work.

---

> ### Author Response · Authors · 2025-11-17
> **Response to Reviewer 1s5C (Supplement to the response to Question 4, as well as the responses to Questions 5 and 6)**
>
> ***(h) Developing simpler and more mathematically coherent variants.*** (***The original comments of the reviewers:*** do something simpler and different in the same vein that is a mathematically coherent method, because the current approach is not.)
>
> ***Reply:*** We agree that our current instantiation (e.g., the hand-designed ratio operator and specific modulation form) is heuristic and not the only coherent option. We already include several ablations (e.g., removing the ratio, using raw future segments, naive concatenation, RAFT-style plug-in) that explore simpler alternatives under the same retrieval framework. In future work, we plan to go further by replacing the hand-crafted transformations with learnable, multi-variate library encoders and studying them within a clearer statistical framework. We will highlight this in the “limitations and future work” section, as the reviewer suggests.
>
> In summary, we are grateful for the reviewer's detailed feedback and for the revised score. We acknowledge that our original exposition—especially around “future continuation / future prior” and the information-theoretic framing—was problematic and could be read as contradicting fundamental principles such as the data-processing inequality. We will revise the manuscript to (i) adopt the non-parametric library-feature framing the reviewer suggests, (ii) clarify assumptions and data usage, (iii) tone down theoretical claims, and (iv) explicitly state the current heuristic nature of some design choices and the need for a more principled statistical treatment in future work.
>
> ***Q5. On the conceptual validity of the ratio operator $\mathbf{R}_j $: alignment and phase sensitivity between history and continuation.***
>
> ***Reply***: We agree that, in general, there is no principled reason why a future continuation must be compared element-wise to a history of the same length. In our implementation, we deliberately choose equal-length windows $\mathbf{H}_j, \mathbf{F}_j \in \mathbb{R}^{L \times C}$ for a fixed horizon and construct $\mathbf{R}_j $ as a simple, aligned heuristic that captures a coarse notion of “how the continuation differs from its preceding history” at the same resolution. This design choice was not made explicit in the current draft, which understandably makes the operator appear arbitrary; we will clarify this implementation detail and explicitly acknowledge that the alignment is a modelling assumption rather than a theoretically unique construction.
>
> We also accept that $\mathbf{R}_j $ is sensitive to phase and time alignment, as illustrated by the sinusoidal example. We do not claim that $\mathbf{R}_j $ is a canonical or statistically optimal “growth-rate” representation of the continuation: in the revised manuscript, we will explicitly reposition $\mathbf{R}_j $ as a simple, heuristic relative-change operator used to instantiate a first concrete example of continuation-style features, and list its phase sensitivity as an explicit limitation.
>
>
> ***Q6. On ignoring multivariate cross-correlation in $\mathbf{R}_j $***
>
> ***Reply:***We agree that the current definition of $\mathbf{R}_j $ operates per entry and does not explicitly model cross-correlation between different variables. This is a deliberate simplification: our goal in this paper was to test, in a controlled and reproducible setting, whether any reasonable summary of “how similar histories tend to continue” can help diverse backbones, rather than to propose the most expressive possible multivariate operator.
>
> We will make this simplification explicit in the revised manuscript and clearly state that $\mathbf{R}_j $ does not capture cross-variable dependency structures. We view this as an important limitation and a natural direction for future work: replacing $\mathbf{R}_j $ with learnable multi-variate library encoders that jointly summarize history–continuation patterns across variables, and studying them within a more principled statistical framework (e.g., non-parametric / kernel / neural encoders).

---

> ### Author Response · Authors · 2025-11-17
> **Response to Reviewer 1s5C (Questions 7 and 8)**
>
> ***Q7. On shift sensitivity of $\mathbf{R}_j $and the use of $\varepsilon {\mathop{\rm sign}\nolimits} ({\bf{H}})$ in the denominator***
>
> Reply: We agree that the current ratio definition is not shift-invariant: adding a constant baseline to both $\mathbf{H}_j $ and $\mathbf{F}_j $ will in general change $\mathbf{R}_j $ . This reflects the fact that we are working in the original value space and interpreting “relative change” with respect to the observed baseline, which is indeed a modelling choice rather than an invariant property. We do not claim that $\mathbf{R}_j $ has desirable invariance properties; in the revised manuscript, we will explicitly describe this shift sensitivity as a limitation and avoid language suggesting that $\mathbf{R}_j $ is a canonical growth-rate measure.
>
> Regarding the $\varepsilon {\mathop{\rm sign}\nolimits} ({\bf{H}})$ term, we fully acknowledge that this is a numerical-stability patch to avoid division by zero and to preserve sign consistency, not a principled statistical construction. Our intention was to ensure stable training with a simple closed-form operator; we will clarify this in the method section and refrain from overstating the theoretical significance of this denominator form. Overall, we will present $\mathbf{R}_j $ honestly as a hand-crafted, heuristic modulation term with known drawbacks (including shift sensitivity and the need for such a patch), rather than as a theoretically well-founded statistic.
>
>
> ***Q8. On the mathematical coherence of the modulation step***
>
> ***Reply:*** We appreciate the concern and agree that, taken out of context, the modulation step
> $$ \hat{\mathbf{F}}_q = \mathbf{X}_q + \tilde{\mathbf{R}}_q \odot \mathbf{X}_q$$
> can look arbitrary. Our intention is to use $\tilde{\mathbf{R}}_q$ as a multiplicative modulation of the current input, in the spirit of FiLM- or gating-style mechanisms, where an auxiliary signal re-weights or reshapes the main representation. Concretely, $\mathbf{X}_q $ encodes the current history, while $\tilde{\mathbf{R}}_q$ (aggregated from library transformations) provides a continuation-style bias that amplifies or attenuates parts of $\mathbf{X}_q $ before it enters the backbone.
>
> We do not claim that this particular functional form is unique or theoretically optimal. To address the reviewer’s concern, we have conducted ablations where we replace this modulation with naive concatenation plus a linear layer, and observe consistently worse performance across multiple backbones. In the revision, we will (i) clarify that this step should be viewed as a standard feature-level modulation rather than a deep probabilistic statement about “future”, and (ii) explicitly report the “fusion vs naive concatenation” ablation as empirical evidence that this structured modulation is more effective than simply enlarging the feature vector. We also acknowledge that designing more principled modulation mechanisms (e.g., learnable neural modulators) is an important direction for future work.

---

> > ### Author Response · Authors · 2025-11-17
> > **Response to Reviewer 1s5C (Question 9)**
> >
> > ***Q9. Are improvements simply due to “feeding more data from the library”, and could simpler feature constructions work as well or better?***
> >
> > ***Reply:*** We agree that this is an important question: if the gains were solely due to “feeding more data from the library”, any sufficiently large and naive feature expansion might suffice, and our specific design would not be justified. To address this, we have performed several controlled comparisons under matched retrieval and data conditions:
> >
> > •	Ratio vs “w/o ratio”: Removing the ratio and directly using aligned future segments from the library (“w/o ratio”) already helps compared to not using the library, indicating that exploiting how similar histories continue is indeed useful. However, the ratio-based transformation still yields additional gains on some datasets (e.g., Exchange), suggesting that the particular way we aggregate continuation-style information is not entirely redundant.
> >
> > •	Continuation-style features vs RAFT-style target fusion: Under a strictly matched plug-in setting (same training retrieval library, similarity metric, backbones, and training budgets), we replace our continuation-style feature with RAFT’s retrieved target segments (i.e., directly piping target values from similar histories into the backbone). We observe that this RAFT-style feature injection consistently underperforms KUP-BI, indicating that the improvements are not merely due to “more target/ history values”, but relate to how continuation information is summarized and fused.
> >
> > •	Fusion vs naive concatenation: When we replace our feature-level fusion/gating with naive concatenation of the auxiliary feature and the original input followed by a linear layer, performance consistently degrades. This suggests that simply enlarging the feature vector is not sufficient; the specific gated fusion structure plays an important role.
> >
> > We will highlight these ablations more explicitly in the revised manuscript to show that: (i) the benefits are not solely from feeding more data from the library, and (ii) while our current design is heuristic and not uniquely optimal, it performs better than several simpler, more naive alternatives under the same conditions. At the same time, we fully acknowledge that our feature construction is one point in a broader design space, and exploring simpler and more mathematically coherent alternatives (including learned library encoders and modulators) is an important next step, which we will state clearly as future work.
> >
> > In summary, we have done our best to address your comments point by point, and we genuinely acknowledge that your concerns are both insightful and constructive. Your willingness to engage deeply with our work and to suggest a clearer and more coherent framing is something we truly appreciate.
> >
> > For some of the broader questions you raised (e.g., a full statistical treatment of the non-parametric library feature, more principled learned encoders, and a complete analysis of bias–variance and exchangeability), we hope you can understand that they are difficult to fully resolve within the scope and timeline of a single conference submission. Our current goal is more modest: to draw attention to the idea that the continuations of training histories—rather than only the targets immediately following the input-can be explicitly exploited as a source of auxiliary structure for forecasting, and to provide initial empirical evidence that this design space is worth exploring.
> >
> > In our view, an important role of a conference venue is to surface interesting questions and promising directions that can stimulate further research, and we hope our work can contribute in this spirit. We see this as a starting point for further discussion and development, rather than a final word on the topic. We will revise the manuscript accordingly to better reflect this positioning, to clarify our limitations, and to lay out a clearer path for the more in-depth theoretical and methodological work that would be more appropriate for a longer follow-up paper.

---

> ### Comment · Reviewer_1s5C · 2025-11-17
> **Final comments.**
>
> "First, we apologize for our earlier wording (“it would be unfair to criticize… please consider whether the manuscript has been evaluated with sufficient neutrality”). Our intention was to emphasize that our estimate is constructed analogously to standard forecasting practice, not to question the reviewer’s neutrality. In hindsight, this sentence was poorly phrased and understandably frustrating. We fully accept that the reviewer is evaluating the work in good faith and with the goal of improving its clarity and rigor."
>
> No apology needed. Even were reviewer neutrality being questions, that is fair enough, and I would not take any such question badly. At least it gave me chance to reflect on whether there was any particular unconscious bias involved (beyond my own inherent academic perspective on the field, which I am afraid you are saddled with - after all that is part of the point of peer-review).
>
> "We do not claim that this particular functional form is unique or theoretically optimal."
>
> No but you do wish to claim assent for the method over other methods. Hence it does need to be justified, consistent and non-arbitrary. I do not feel the current approach is, and I think a bit of work and insight is needed to turn this into something that is.
>
> "In our view, an important role of a conference venue is to surface interesting questions and promising directions that can stimulate further research, and we hope our work can contribute in this spirit. We see this as a starting point for further discussion and development, rather than a final word on the topic. We will revise the manuscript accordingly to better reflect this positioning, to clarify our limitations, and to lay out a clearer path for the more in-depth theoretical and methodological work that would be more appropriate for a longer follow-up paper."
>
> I absolutely agree that is the point of conferences, and  indeed that in my mind is also as much the point of conference publication. Nothing is ever the final word.
>
> I also realise that the authors would love to get the chance to present this paper to this audience. However I am afraid I am not supportive of that - this is a big conference, and this paper at the moment has too many inconsistencies and difficulties and would confuse as much as it would prompt questions and directions. I think the authors would do well to re-derive a different but related method with a more meaningfully principled and grounded algorithm, and reposition the paper accordingly, tightening the writing and the experimental analysis. To do that properly will take more time than mere post-review revisions. I am hoping that my comments might help  in that process, if not then I apologise. Disappointing as my position might seem, the field does not need more papers. It needs papers written more excellently and doing more excellent things. Hopefully this work may lead to such a paper in the future.  I look forward to seeing it.
>
> My apologies - this will be the last of my comments and changes - I have six other papers reviews to handle too, as well as the day job :). All the best with future iterations of this paper.

---

### Official Review · Reviewer_qV9Q · 2025-10-30

**Soundness:** 3
**Presentation:** 3
**Contribution:** 2
**Rating:** 2
**Confidence:** 4

**Summary:**

The paper presents a novel algorithm of using future priors developed independently to improve time series forecasting by leveraging both the historical values and the future priors. The paper shows improvements in the forecasts based on this methodology across standard benchmark datasets within the time series foundation models community using three forecasting models, namely, PatchTST, DLinear and TimesNet.

**Strengths:**

Paper presents a novel algorithm by reframing the forecasting problem as  an interpolation problem rather than the conventional extrapolation problem to improve time series forecasting using future priors across a wide range of benchmark datasets.

Algorithm also allows for any other future prior to be utilized and plugged into the algorithm.
Significant gains in performance have been shown for DLinear algorithm and modest gains for PatchTST and TimesNet.

**Weaknesses:**

It is not clear what are the advantages of this method as compared to existing methods mentioned in the literature review by the authors in Section 2.2.

I think a comparison with state of the art methods is missing. I think comparing the algorithm mentioned in this paper with RAFT model in Section 2.2 or other models in Section 2.2 will illustrate the benefits of this work better (if any).

State of the art models can be improved. TTM and Chronos are probably more standard models employed right now as compared to PatchTST.

**Questions:**

As stated in the weaknesses, I think the authors can expand on this section to make the applicability of this method more clear with reference to prior art in Section 2.2.

1. Where would this method work better than the existing methods?
2. Are there any computational benefits compared to the existing methods?
3. How does this method compare with other state-of-the-art methods like RAFT?

---

> ### Author Response · Authors · 2025-11-14
> **Response to Question 1 (Where would this method work better than the existing methods?)**
>
> ***Response to Question 1 (Where would this method work better than the existing methods?)***
>
> We thank the reviewer for the efforts reviewing our work and providing helpful advice for our proposed method.
>
> First, we would like to clarify the differences between existing methods and the proposed KUP-BI, which will help better contextualize the advantages of our approach over prior work.
>
> ***Existing methods.*** Retrieval-based approaches (e.g., Han-RAFT (2025) and Tire-RAF (2024)) typically retrieve the ***target*** segment in the chain “***history*** → ***target***→ ***future continuation***”, and then either concatenate the ***target*** segment to the input or fuse it after making separate predictions. In essence, they remain single-stream history-to-future extrapolation.
>
> ***Our method (KUP-BI)***. We learn, from the training library, the relative-change ratio between a ***history*** and its ***future continuation*** in the same chain, and use it to construct a proxy ***future continuation*** $\hat F$ for the current input. We then encode this proxy in parallel with the current stream and perform gated convex fusion. This can be viewed as a two-stage conditioning augmentation, while the training objective remains the standard MSE loss. ***Our focus is on how to construct a future continuation; retrieval is merely the means to obtain comparable histories. Therefore, our work and existing retrieval-based methods are different, and they can also be combined to further improve a given backbone.***
>
> Han et al. Retrieval Augmented Time Series Forecasting, 2025, pub in ICML.
>
> Tire et al. Retrieval Augmented Time Series Forecasting, 2024, pub in arXiv.
>
> ***Why is our method better?***
>
> In short, KUP-BI injects a horizon-aligned, ratio-conditioned proxy of the ***future continuation*** and fuses it with the current stream without extending the sequence, yielding efficient and stable gains while preserving the time axis. The advantages of the proposed method are as follows.
>
> 1.	Using future continuation is better than only using the target. Traditional time-series models already learn the mapping  “***history*** → ***target***→ ***future continuation***” during training, so retrieving the ***target*** typically just reinforces patterns the backbone already handles, yielding limited marginal gains. In contrast, ***future continuation*** provides additional structural information about how the series evolves after the ***target***, which is precisely where backbones are relatively weaker—especially on long horizons.
>
> 2.	Avoiding temporal index alignment risk. Directly concatenating a ***target*** segment from another day into the current sequence is equivalent to inserting an external time index into the current timeline, which makes the model more susceptible to amplitude/seasonality strength differences and slight phase misalignment. For models that rely on timestamps to capture periodicity, this can cause them to lock onto spurious or incorrect seasonal cycles. In contrast, our method injects the ***future continuation*** structural information without altering the time axis, thereby avoiding this risk.
>
> 3.	Robust fusion and engineering simplicity. “Dual-stream + gated convex fusion (γ, α)” keeps the current stream dominant and guarantees the basic performance of the prediction model when the estimated ***future continuation***  is weak. The retrieval library is built from the training set and kept frozen (preventing leakage), and no backbone attention modification is required.
>
> 4.	Significant improvements for light backbones. When the backbone is lighter (e.g., DLinear), cross-series ratio structure provides information that the backbone does not reliably learn on its own, leading to significant performance improvements for light backbones.
>
> In addition, we compare KUP-BI with the plugin version of Han-RAFT under strictly matched conditions: identical backbones (PatchTST, TimesNet, xPatch, DLinear), data splits, optimizers, and training budgets. The retrieval library is built only from the training set to avoid leakage. Han-RAFT follows its original method: for each query history, it retrieves the targets of similar training histories, computes softmax weights over the similarities, and forms a weighted sum (“target prototype”); the prototype is then passed through a linear projection and convexly combined with the backbone’s prediction to produce the final output.
>
> ***As shown in Table 1, for the three backbones (DLinear, xPatch, TimesNet), KUP-BI yields larger gains than RAFT on both ETTh1 and ETTh2. For PatchTST, KUP-BI outperforms RAFT on ETTh2 and is comparable on ETTh1. Overall, KUP-BI is stronger.***

---

> ### Author Response · Authors · 2025-11-14
> **Response to Question 1 (Where would this method work better than the existing methods?)**
>
> ***Response to Question 1 (Where would this method work better than the existing methods?)***
>
>
> **Table 1.** KUP-BI vs. RAFT across four backbones on ETTh1/ETTh2 and four horizons (MSE/MAE ↓). Best results in **bold**.
> | Model | PatchTST+ KUP-BI | PatchTST+ RAFT | TimesNet+ KUP-BI | TimesNet+ RAFT | DLinear+ KUP-BI | DLinear+ RAFT | xPatch+ KUP-BI | xPatch+ RAFT |
> |-------|------------------|----------------|------------------|----------------|-----------------|---------------|----------------|--------------|
> | Metric | MSE & MAE | MSE & MAE | MSE & MAE | MSE & MAE | MSE & MAE | MSE & MAE | MSE & MAE | MSE & MAE |
> | ETTh1 | 0.409 & 0.425 | **0.406 & 0.423** | **0.453 & 0.453** | 0.480 & 0.469 | **0.425 & 0.437** | 0.488 & 0.483 | **0.409 & 0.422** | 0.415 & 0.426 |
> | ETTh2 | ***0.327 & 0.376*** | 0.334 & 0.382 | ***0.396 & 0.414*** | 0.424 & 0.433 | ***0.394 & 0.426*** | 0.542 & 0.495 | ***0.338 & 0.381*** | 0.348 & 0.390 |

---

> ### Author Response · Authors · 2025-11-14
> **Response to Question 2 (Are there any computational benefits compared to the existing methods?)**
>
> ***Response to Question 2 (Are there any computational benefits compared to the existing methods?)***
>
> Thanks for the suggestion. Overall, KUP-BI's computational efficiency is close to that of Han-RAFT and slightly better than RAF. We decompose the analysis into two parts: (i)***retrieval cost***and (ii) ***model forward cost***.
>
> ***Retrieval stage (identical)***. All methods perform Top-k retrieval from a frozen training library, with the same implementation and cost.
> ***Model forward stage (main difference)***. KUP-BI and Han-RAFT both ultimately produce an input of length $L$ (the same as the original input) and both use a lightweight weighted fusion. The difference is that KUP-BI adds patching and a linear projection, whereas Han-RAFT adds a linear projection for the retrieved targets. These differences have only a very small impact on efficiency, so the two methods are very close. Unlike KUP-BI and Han-RAFT, RAF produces a new input of length $L+H$, where $H$ is the target-window length. For Transformer backbones, due to attention, the computational cost is approximately $ \mathcal{O}( L^2)$ for KUP-BI and Han-RAFT, versus $\mathcal{O}({(L+H}^2)$ for RAF. For linear/convolutional backbones, the differences are small. Therefore, KUP-BI and Han-RAFT are close in efficiency, while RAF is slightly less efficient than the other two.
>
> We report the impact of different plug-ins on two backbones across two datasets (ETTh1 and ETTh2). Table 2 provides the average epoch time for each model at predicted length H=96. To ensure fairness, all model hyperparameters are kept identical. From Table 2 we observe that on DLinear the three methods are very close; on PatchTST, KUP-BI and Han-RAFT remain close, while RAF shows a slightly higher epoch time than the first two.
>
> **Table 2.** Average epoch time (seconds) on **ETTh1 and ETTh2** under the same settings.
>
> | Model | PatchTST+ KUP-BI | PatchTST+ RAFT | PatchTST+RAF | DLinear+ KUP-BI | DLinear+ RAFT | DLinear+RAF |
> |-------|------------------|----------------|----------|------------------|----------------|---------|
> | ETTh1 |   1.914          | 1.941          | 2.941    | 1.193            | 1.207          | 1.332   |
> | ETTh2 |    1.622         | 1.590          | 2.228    | 1.570            | 1.482          | 1.553   |

---

> > ### Author Response · Authors · 2025-11-14
> > **Response to Question 3 (How does this method compare with other state-of-the-art methods like RAFT?)**
> >
> > ***Response to Question 3 (How does this method compare with other state-of-the-art methods like RAFT?)***
> >
> >
> > Thanks for the helpful suggestion. After carefully investigating the related literature, to the best of our knowledge, Han-RAFT is currently the latest and most relevant published work to ours. We also carefully examine the related work in the manuscript of Han-RAFT. It does not discuss other similar approaches that have been already published, likely because this specific area is still relatively unexplored. We try to fill the gap in this work.
> >
> > As mentioned above, we mainly take Han-RAFT as the representative for comparison. Basically, our method is a plugged-in tool but RAFT is essentially a forecasting method that cannot be directly used as a plug-in. The conceptual differences between these two methods have been discussed in our response to ***Q1***. To experimentally compare them fairly, we follow Han-RAFT’s retrieval pipeline, i.e., we retrieve the ***target*** segments corresponding to historical segments similar to the current input. We feed the original input into the backbone to obtain a prediction. In Han-RAFT, the retrieved ***target*** segments are passed through a linear layer; we keep this operation. Finally, we obtain the model’s prediction by weighted fusion of this output with the prediction produced from the original input. Under this setting, Han-RAFT uses the same hyperparameters as KUP-BI, enabling a fair comparison. The results for Han-RAFT and KUP-BI are summarized in Table 1. As shown in Table 1, ***KUP-BI provides greater benefits to the backbone models than Han-RAFT. Also, KUP-BI is comparable to Han-RAFT in computational efficiency, as shown in Table 2.***

---

> > > ### Author Response · Authors · 2025-11-14
> > > **Response to Question 4 (State of the art models can be improved. TTM and Chronos are probably more standard models employed right now as compared to PatchTST)**
> > >
> > > ***Response to Question 4 (State of the art models can be improved. TTM and Chronos are probably more standard models employed right now as compared to PatchTST)***
> > >
> > > We thank the reviewer for this valuable suggestion. In fact, KUP-BI can be combined with Chronos/TTM in the same way as in Tire-RAF (2024). However, we did not include TTM/Chronos in our current experiments in order to maintain fairness. To isolate KUP-BI's contribution, TTM/Chronos must be fully frozen, and only a tiny, unified gating/fusion head should be trained, while keeping the same retrieval index, Top-k, similarity, and temperature. Many available implementations differ in preprocessing / time encoding / distilled priors from our baseline suite. Following your suggestion, we have conducted additional experiments on a very recent SOTA backbone, namely xPatch (2025). ***As can be seen from Table 3, after introducing KUP-BI into xPatch, the performance of the model has been steadily improved.***
> > >
> > > **Table 3.** Long-term prediction results of xPatch and xPatch+KUP-BI on six public datasets. The better results are highlighted in bold.
> > >
> > > | Model | xPatch (2025) | xPatch+KUP-BI |
> > > |-------|--------|----------------|
> > > | Metric | MSE & MAE | MSE & MAE |
> > > | ETTh1 | 0.444 & 0.438 | **0.409 & 0.422** |
> > > | ETTh2 | 0.342 & 0.383 | **0.338 & 0.381** |
> > > | ETTm1 | 0.352 & 0.372 | **0.350 & 0.372** |
> > > | ETTm2 | 0.252 & 0.308 | **0.250 & 0.308** |
> > > | ILI | 1.383 & 0.718 | **1.365 & 0.712** |
> > > | Exchange | 0.364 & 0.403 | **0.359 & 0.402** |
> > > | Improvement | 2.173% (MSE) | **0.866% (MAE)** |

---

> > > > ### Author Response · Authors · 2025-11-16
> > > > **Response to Question 5 (What are the differences between us and the existing methods?)**
> > > >
> > > > ***Response to Question 5 (What are the differences between us and the existing methods?)***
> > > >
> > > >
> > > > In summary, exogenous-variable and retrieval-augmented methods primarily enrich the mapping along the history–target link in the natural chain ``***history*** (model input)–***target*** (ground-truth output)–f***uture continuation***''.
> > > > By contrast, KUP-BI explicitly constructs a ***future continuation*** stream that lies beyond the prediction horizon and treats it as an additional input stream to the backbone. Its novelty lies in introducing and exploiting this future-oriented stream as an auxiliary feature signal for conditioning the prediction, rather than in any particular mechanism for constructing it.

---

### Official Review · Reviewer_yB2k · 2025-10-31

**Soundness:** 3
**Presentation:** 2
**Contribution:** 2
**Rating:** 6
**Confidence:** 3

**Summary:**

This paper introduces KUP-BI (Knowledge Utilization Paradigm with Bidirectional Inference), a framework that augments standard unidirectional time-series forecasting models with an auxiliary future-prior stream. Instead of predicting solely from historical data (past → future), KUP-BI retrieves approximate future priors from a library of historical patterns (constructed as history–target–future chains). These priors represent structural cues about plausible future trajectories and are fused with the current input’s representation via a lightweight gated module. The design reframes forecasting from a purely extrapolative problem into an interpolation-like one, using approximate structural knowledge from correlated historical patterns to stabilize predictions.
Empirical results on six datasets (ETTh1/2, ETTm1/2, Exchange, and ILI) with three backbones (DLinear, TimesNet, PatchTST) show consistent improvements in MSE/MAE, particularly for lightweight models (e.g., DLinear). The authors also provide theoretical motivation (an interpolation vs. extrapolation error bound under Lipschitz continuity) and detailed ablations on retrieval quality, fusion strategies, and hyperparameters.

**Strengths:**

The conceptual novelty lies in explicitly incorporating an estimated “future prior”—learned from training histories that resemble the current input—as a structural anchor to guide prediction. This moves beyond existing retrieval-augmented or exogenous-enhanced methods, which either concatenate retrieved outputs or use external signals. The bidirectional reasoning idea—combining historical and prospective cues—is fresh and supported by a solid theoretical argument that interpolation is less error-prone than extrapolation for Lipschitz functions.

While related to retrieval-augmented forecasting (RAFT, TS-RAG), future prior modeling as a separate learnable stream with a ratio-based representation (rather than directly fusing retrieved outputs) appears original. The method’s simplicity and plug-and-play nature increase its potential impact—especially since it can attach to diverse backbones with negligible overhead.

- The ratio-style operator ($R = (F-H)/(H+\epsilon,\text{sign}(H))$) is clearly defined, and the retrieval and softmax-weighted fusion mechanisms are mathematically specified.


- The Lipschitz interpolation theorem is formally derived in Appendix A and conceptually motivates the approach. While idealized, it reinforces the intuition that adding a bounded-fidelity future anchor reduces variance.


- The gated fusion module and harmonic residual design (convex combination controlled by α and per-channel gate γ) are reasonable and stabilizing.


- The experiments seem to be methodically conducted with non-leaking retrieval libraries (constructed only from training data).

**Weaknesses:**

I believe

- the retrieval-based prior depends on correlation measures that may fail under phase shifts or abrupt regime changes. Although the authors acknowledge this, no quantitative robustness analysis is included.

- the “ratio operator” assumes relative amplitude continuity between history and future, which might not hold in nonstationary series (financial or event-driven).


While results are statistically consistent, the improvements are modest (1–7% MSE reduction). It would help to report statistical significance or confidence intervals. Also, the approach may implicitly leak periodic information if future segments overlap with nearby training windows—clarifying how window offsets are handled would strengthen reproducibility, given that code is not shared yet.

**Questions:**

- Can you quantify retrieval precision (e.g., correlation between estimated and true future segments) to better understand when the prior helps?


- How does KUP-BI perform on non-periodic or abrupt-shift datasets (e.g., stock tick data)?


- Could a learned retriever (e.g., embedding-based) outperform correlation-based matching?


- The ratio operator assumes elementwise alignment—could phase-shift alignment or dynamic time warping improve the prior’s fidelity?


- Can you include some runtime comparisons (ms per batch) to substantiate claims of negligible overhead?

---

> ### Author Response · Authors · 2025-11-19
> **Response to Questions 1 and 2**
>
> ***Q1: Can you quantify retrieval precision (e.g., correlation between estimated and true future segments) to better understand when the prior helps?***
>
> We thank the reviewer for this helpful suggestion.
> Following your advice, we conducted a quantitative evaluation of the constructed auxiliary sequence on the test set, using DLinear as the backbone.
> (For clarity, what we previously referred to as a “future prior” is now termed a ***continuation-style auxiliary segment*** to avoid ambiguity.)
>
> For each sample, we align the constructed continuation-style auxiliary segment (length $L$, channels $C$) with its corresponding ground-truth post-target segment (the $L$ steps immediately after the target window $T$), flatten both into vectors, and compute the Pearson correlation coefficient $r$ and the mean squared error of the auxiliary ($MSE_{\text{aux}}$).
>
>
> Table 1 reports both the forecasting metrics (MSE and MAE) and the auxiliary-quality metrics ($\operatorname{mean}|r|$, $MSE_{\text{aux}}$) on ETTh1/2 and ETTm1/2. To ensure fairness and reproducibility, all hyperparameters (Top-$k$, $\tau$, $L$, $T$, etc.) follow the main experiments, and the retrieval library is built only from the training set and kept frozen during evaluation to avoid data leakage; the ground-truth post-target segments are used here solely as a diagnostic signal to assess the quality of the auxiliary sequence, not as an input to the model.
>
> From Table 1, we observe a clear trend: KUP-BI’s gains are positively associated with the quality of the continuation-style auxiliary sequence. Within datasets of the same type and temporal granularity, a higher-quality auxiliary (larger $\operatorname{mean}|r|$, smaller $MSE_{\text{aux}}$) tends to yield larger improvements. For example, between ETTh1 and ETTh2, ETTh2 has both higher correlation and lower MSE than ETTh1; accordingly, KUP-BI achieves larger error reductions on ETTh2. Overall, the closer the auxiliary sequence is to the true post-target continuation, the more reliably KUP-BI reduces forecasting error; in more irregular or high-noise settings, the auxiliary becomes less informative, and the benefit correspondingly diminishes.
>
> ***Table 1. Effect of Retrieved Continuation-Style Auxiliary Segment on Forecasting Accuracy (DLinear, Test Set)***
> | Model | DLinear (MSE) | DLinear (MAE) | DLinear+KUP-BI (MSE) | DLinear+KUP-BI (MAE) | Improvement (MSE) | Improvement (MAE) | Auxiliary-quality mean $\lvert r \rvert$ | Auxiliary-quality $\mathrm{MSE}_{pr}$ |
> |-------|---------------|----------------|------------------------|------------------------|--------------------|--------------------|--------------------------------------|----------------------------------------|
> | **Metric** | **MSE** | **MAE** | **MSE** | **MAE** | **MSE** | **MAE** | mean $\lvert r \rvert$ | $\mathrm{MSE}_{pr}$ |
> | ETTh1 | 0.445 | 0.454 | **0.425** | **0.437** | 4.403% | 3.848% | 0.588 | 0.803 |
> | ETTh2 | 0.469 | 0.463 | **0.394** | **0.426** | 15.887% | 8.096% | 0.883 | 0.342 |
> | ETTm1 | 0.359 | 0.381 | **0.358** | **0.380** | 0.181% | 0.176% | 0.471 | 1.252 |
> | ETTm2 | 0.283 | 0.345 | **0.266** | **0.330** | 5.930% | 4.179% | 0.926 | 0.222 |
>
> ***Q2: How does KUP-BI perform on non-periodic or abrupt-shift datasets (e.g., stock tick data)?***
>
> Response. Thank you for the insightful question. While we do not include stock tick data in our current benchmark, we do evaluate KUP-BI on datasets that are much less periodic and more irregular than ETTh1, such as ETTm1 and Exchange. On these datasets, KUP-BI still brings consistent but smaller gains over the backbones, which is consistent with the analysis in Table 1: the more irregular ETTm1 exhibits lower auxiliary quality (smaller $\operatorname{mean}|r|$ and larger $MSE_{\text{aux}}$) and correspondingly only marginal improvement.
>
> Intuitively, in highly non-periodic or abrupt-shift settings (e.g., high-frequency financial ticks), the train-only library is less likely to contain many truly analogous “history-post-target continuation” chains for a given query, so the constructed continuation-style auxiliary sequence becomes weaker and noisier. In such cases, KUP-BI is designed to behave conservatively: the gated convex fusion (parameterized by $\gamma$ and $\alpha$) is learned end-to-end with the forecasting loss, and implicitly down-weights the auxiliary stream when it does not help. Empirically, across all datasets we tested (including the more irregular ones), we do not observe systematic degradation relative to the backbone; instead, the model tends to return to the main information when the auxiliary is uninformative.

---

> ### Author Response · Authors · 2025-11-19
> **Responses to Questions 3 and 4**
>
> ***Q3: Could a learned retriever (e.g., embedding-based) outperform correlation-based matching?***
>
>
> Response. We appreciate this insightful question. In the current paper, we deliberately adopt a simple, non-parametric correlation-based retriever for two reasons: (i) it keeps the additional computational and implementation overhead minimal and fully transparent, and (ii) it allows us to cleanly isolate the contribution of the proposed continuation-style auxiliary construction and gated fusion, without mixing it up with gains from a more complex retrieval module.
> Conceptually, KUP-BI does not depend on any particular retrieval mechanism. It only needs a similarity score $s(\mathbf{X}, \mathbf{H}_j)$ between the current input $\mathbf{X}$ and each library history $\mathbf{H}_j$, so the correlation-based retriever used in this paper can be replaced by any embedding-based or learned retriever. For example, one could learn an encoder $\varphi(\cdot)$ and retrieve neighbours in the embedding space using dot-product or cosine similarity, trained with a suitable objective on “history–post-target continuation” chains.
>
> We therefore see ***learned, embedding-based retrievers as a promising way to further enhance KUP-BI***: a stronger retriever would directly improve the quality of the constructed continuation-style auxiliary sequence, and thus could potentially lead to additional gains. A systematic study of such learned retrievers, and of their trade-offs between accuracy and overhead, is beyond the scope of the current submission, but we will explicitly mention this extension as an important direction for future work in the revised manuscript.
>
> ***Q4: The ratio operator assumes elementwise alignment—could phase-shift alignment or dynamic time warping improve the prior’s fidelity?***
>
> Response.
> We thank the reviewer for this valuable suggestion. You are correct that our current ratio operator implicitly assumes elementwise alignment: we compute the ratio at the same time index on both the history and the post-target continuation within each training chain, and then apply the fused ratio elementwise to the current input. When there is a noticeable phase shift between two similar patterns, this elementwise construction can indeed dilute the quality of the constructed continuation-style auxiliary sequence. This is the “phase misalignment” issue that we briefly mentioned as a limitation in the conclusion, and we agree that it deserves a more explicit discussion.
>
> A natural extension is to explicitly align the retrieved segments before computing the ratio. For example:
> (1) one can search over a small lag window and choose the phase shift that maximizes cross-correlation, then align the retrieved history-post-target continuation chain to this lag and apply the same ratio operator;
> (2) alternatively, one could adopt a lightweight DTW-style alignment to handle local time misalignment before forming the ratio;
> (3) or define the ratio in the frequency domain (e.g., based on amplitude spectra), which is inherently more robust to phase shifts.
>
> In all of these cases, the overall KUP-BI framework remains unchanged, and only the local definition of the ratio operator becomes phase-aware. In the current paper, we intentionally use the simplest elementwise instantiation to keep the method transparent and computationally cheap, while relying on the learned gated fusion to down-weight auxiliary streams that do not help the forecasting loss. We will add a dedicated paragraph in the revised manuscript to clearly state this alignment assumption and to discuss phase-aware ratio variants as a promising direction for further improving the quality of the continuation-style auxiliary sequence.

---

> > ### Author Response · Authors · 2025-11-19
> > **Response to Question 5 (Can you include some runtime comparisons (ms per batch) to substantiate claims of negligible overhead?)**
> >
> > ***Q5: Can you include some runtime comparisons (ms per batch) to substantiate claims of negligible overhead?***
> >
> >
> > Thank you for the suggestion.
> >
> > We measured the training time per batch on ETTh1 and ETTh2 for four backbones (PatchTST, DLinear, TimesNet, xPatch) and four prediction horizons (96, 192, 336, 720), using the same batch size and a single GPU. The detailed numbers are reported in Table~12. Averaged over the four horizons on ETTh1, we obtain:
> > 1) PatchTST: from 11.6 ms/batch to 15.4 ms/batch (about 1.3 times, +3.8 ms);
> > 2) DLinear: from 2.28 ms/batch to 5.08 ms/batch (about 2.2 times, +2.8 ms);
> > 3) TimesNet: from 36.8 ms/batch to 77.4 ms/batch (about 2.1 times);
> > 4) xPatch: from 11.0 ms/batch to 11.0 ms/batch (essentially unchanged).
> >
> > On ETTh2, we observe similar trends: PatchTST and xPatch show at most about 0–20\% extra time (within 3–4 ms per batch), DLinear shows a larger relative factor (about 1.4 times) but the absolute overhead remains small (around 1–3 ms per batch), and TimesNet shows roughly a 2 times increase.
> >
> > Overall, these results show that for typical patch-based backbones (PatchTST, xPatch), KUP-BI behaves as a lightweight plug-in with only a few additional milliseconds per batch. For very small or very heavy models (DLinear, TimesNet), the relative overhead can be higher, mainly because our current implementation uses exact correlation-based retrieval over a large library. This cost is not inherent to KUP-BI itself and can be further reduced with standard engineering optimizations.
> >
> > In the revised manuscript, we present the corresponding content in Table 12 and modify "negligible" in the original text to "small additional overhead".
> >
> >
> >
> > Table 2.Training-time cost (ms/batch) on ETTh1/2 for four backbones.
> > | Dataset | Pred | PatchTST Ori | PatchTST + KUP-BI | DLinear Ori | DLinear + KUP-BI | TimesNet Ori | TimesNet + KUP-BI | xPatch Ori | xPatch + KUP-BI |
> > |---------|------|--------------|-------------------|-------------|------------------|--------------|-------------------|------------|-----------------|
> > | ETTh1   | 96   | 11.15        | 14.79             | 2.16        | 6.23             | 27.97        | 36.76             | 9.57       | 9.30            |
> > | ETTh1   | 192  | 11.06        | 15.12             | 2.29        | 4.21             | 30.62        | 51.56             | 10.52      | 9.69            |
> > | ETTh1   | 336  | 11.74        | 15.82             | 2.14        | 4.77             | 35.00        | 70.39             | 11.51      | 11.64           |
> > | ETTh1   | 720  | 12.47        | 15.95             | 2.53        | 5.12             | 53.54        | 151.01            | 12.36      | 13.17           |
> > | ETTh1   | Avg  | 11.61        | 15.42             | 2.28        | 5.08             | 36.78        | 77.43            | 10.99      | 10.95           |
> > | ETTh2   | 96   | 11.06        | 10.09             | 1.92        | 2.76             | 29.25        | 51.67             | 6.35       | 7.75            |
> > | ETTh2   | 192  | 10.92        | 10.14             | 2.03        | 2.87             | 33.29        | 61.38             | 6.51       | 8.12            |
> > | ETTh2   | 336  | 11.47        | 10.28             | 2.06        | 2.90             | 38.66        | 75.20             | 6.64       | 7.88            |
> > | ETTh2   | 720  | 12.21        | 10.64             | 2.22        | 3.01             | 65.41        | 165.50            | 7.25       | 8.42            |
> > | ETTh2   | Avg  | 11.42        | 10.29             | 2.06        | 2.89             | 41.65        | 88.44            | 6.69       | 8.04            |

---

### Author Response · Authors · 2025-11-30
**Summary of Revisions and Responses for the Area Chair (Part 1)**

***Summary of Revisions and Responses for the Area Chair***

We sincerely thank the area chair and the reviewers for their time. For the area chair’s convenience, we summarize the main clarifications and revisions made in response to the reviews.

Our contributions to the community are as follows:

***Contribution:*** Our work revisits the standard one-way formulation of time-series forecasting (history→target) and identifies an overlooked source of information: the post-target continuation in training trajectories, i.e., how sequences continue after the usual prediction horizon. In supervised forecasting datasets, each training example naturally forms a chain: ***history (model input)*** → ***target (ground-truth output)*** → ***post-target continuation*** (future beyond the prediction horizon). Standard forecasters only use the first two parts of this chain. In our work, we explicitly make use of the third part, the post-target continuation segment. We implement this idea in KUP-BI, a simple plug-in module that can be attached to standard backbones. Across multiple backbones and benchmarks, exploiting this continuation source yields consistent error reductions with small overhead, suggesting a new family of methods beyond pure extrapolation from history.

According to the reviewers' comments, our work has been recognized for ***its conceptual novelty, its new perspective on time-series forecasting, and its empirical effectiveness across multiple backbones and benchmarks***. For clarity, we highlight a few representative remarks below.

***Reviewer yB2k:*** The conceptual novelty lies in explicitly incorporating an estimated “future prior”—learned from training histories that resemble the current input—as a structural anchor to guide prediction. This moves beyond existing retrieval-augmented or exogenous-enhanced methods….. (***Strengths, first review***).

***Reviewer qV9Q***: Paper presents a novel algorithm by reframing the forecasting problem as an interpolation problem rather than the conventional extrapolation problem to improve time series forecasting using future priors across a wide range of benchmark datasets. (***Strengths, first review***).

***Reviewer 1s5C***: I will up my score a bit, as I now see that there is something being done here that might potentially make some sense, simply because it is not really doing a predicted future, but just computing a feature vector from the library that is used during prediction. (***comment titled “Continued”, 17 Nov 2025, 03:56***).

At the same time, the reviewers also pointed out important weaknesses, mainly:

1.	The experimental evaluation was incomplete. In particular, the original version did not include explicit efficiency (runtime) measurements and did not compare against the most recent state-of-the-art methods.

2.	The original version did not spell out some key assumptions and data-usage details in a fully explicit way. This included the mild conditional stationarity assumption (similar histories lead to similar futures) and the fact that the retrieval library is built only from the training split, which led to concerns about possible data leakage and the fairness of comparisons.

3.	The non-parametric retrieval mechanism was presented in a mainly empirical way, with limited theoretical discussion, and the feature-construction pipeline initially lacked detailed ablation studies. This made some stages of the design look more hand-crafted or arbitrary than intended.

4.	Some descriptions were potentially misleading. For example, the term “Retrieved Future Prior” could cause readers to misunderstand our method as mixing prior and posterior distributions or even altering basic probabilistic foundations.

We have carefully read the reviewers’ comments and revised the manuscript accordingly. Overall, we believe that the revised paper has been substantially improved and we hope it will satisfy the reviewers.

We also note that, following the suggestion from Reviewer 1s5C, we have renamed “***Retrieved Future Prior***” as “***Retrieved Continuation-style Auxiliary.***”

---

> ### Author Response · Authors · 2025-11-30
> **Summary of Revisions and Responses for the Area Chair(Part 2)**
>
> ***Response to Reviewer yB2k (Rating: 6)***
>
> ***Q1: Can you quantify retrieval precision (e.g., correlation between estimated and true future segments) to better understand when the prior helps?***
>
> Response: We thank the reviewer for raising this question. To address this request, the revised version includes a new analysis in ***Appendix H on page 19 and Table 13 on page 21***. There, we compute the correlation and mean squared error (MSE) between the estimated and true future segments and show that higher retrieval precision (higher correlation and lower MSE) leads to larger improvements.
>
> ***Q2: How does KUP-BI perform on non-periodic or abrupt-shift datasets (e.g., stock tick data)?***
>
> Response: Thanks for your comment. Although we did not include stock tick data in our experiments, the revised version provides indirect evidence from ***Table 13 on page 21***. In that table, we report results on two datasets: ETTh1, which exhibits clearer seasonal patterns, and ETTm1, which is more irregular and less periodic. The proposed KUP-BI brings smaller gains on ETTm1 than on ETTh1, suggesting that the method is less effective when the data are non-periodic or contain abrupt shifts.
>
> ***Q3: Could a learned retriever (e.g., embedding-based) outperform correlation-based matching?***
>
> Response: We thank the reviewer for raising this question. In principle, a learned retriever could be more powerful than our simple correlation-based matching. However, the main goal of this paper is to highlight a new information source – the continuation segment in training trajectories – and to test its value in a very simple setting. For this reason, we deliberately use a basic, non-learned retriever. Designing and training a stronger embedding-based retriever on top of this idea is left for future work.
>
> ***Q4: The ratio operator assumes elementwise alignment—could phase-shift alignment or dynamic time warping (DTW) improve the fidelity of the retrieved continuation-style auxiliary?***
>
> Response: This is a very insightful suggestion. Phase alignment (or a simple dynamic time warping step) before applying the ratio operator could indeed improve the accuracy of the retrieved continuation-style auxiliary and make the proposed KUP-BI more stable. However, as mentioned in the previous answer, the main goal of this paper is to draw attention to a new information source and study it in a very simple setting. For this reason, we intentionally keep the ratio operator basic and do not add extra alignment modules. Incorporating phase-shift alignment or DTW-style matching is a promising extension that we leave for future work.
>
> ***Q5: Can you include some runtime comparisons (ms per batch) to substantiate claims of negligible overhead?***
>
> Response: We thank the reviewer for this comment. We have added runtime comparisons in the revised version. As shown in ***Table 12 on page 21***, introducing the proposed KUP-BI only brings a very small increase in running time.

---

> ### Author Response · Authors · 2025-11-30
> **Summary of Revisions and Responses for the Area Chair (Part 3)**
>
> ***Response to Reviewer qV9Q (Rating: 2)***
>
> ***Q1. Where would this method work better than the existing methods?***
>
> Response: Thanks for your comment. To our knowledge, there is currently no existing method that studies using the post-target continuation as an information source in the same way as we do. The closest works are retrieval-based approaches such as RAF (Tire et al. (2024)). ***The proposed KUP-BI works better than RAF in settings where the backbone model uses explicit time-stamp information (for example, TimeEmb (Xia et al., 2025)).*** In such settings, KUP-BI is more suitable than RAF because it keeps the original time axis unchanged, whereas RAF concatenates retrieved sequences and shifts the time indices.
>
> ***Q2. Are there any computational benefits compared to the existing methods?***
>
> Response: We thank the reviewer for this comment. Here, we analyze two recent retrieval-based works, RAF (Tire et al., 2024) and RAFT (Han et al., 2025). Overall, the proposed KUP-BI has computational efficiency that is close to that of RAFT and slightly better than RAF.
>
> We decompose the analysis into two parts: (i) retrieval cost and (ii) model forward cost.
> Retrieval stage (identical). All methods perform Top-k retrieval from a frozen training library, with the same implementation and cost.
> Model forward stage (main difference). The proposed KUP-BI and RAFT both ultimately produce an input of length L (the same as the original input) and both use a lightweight weighted fusion. The difference is that KUP-BI adds patching and a linear projection, whereas RAFT adds a linear projection for the retrieved targets. These differences have only a very small impact on efficiency, so the two methods are very close. Unlike KUP-BI and RAFT, RAF produces a new input of length L+H, where H is the target-window length. For Transformer backbones, due to attention, the computational cost is approximately O(L2) for KUP-BI and RAFT, versus O((L+H)2) for RAF. For linear/convolutional backbones, the differences are small. Therefore, KUP-BI and RAFT are close in efficiency, while RAF is slightly less efficient than the other two.
>
>
>
> ***Q3. How does this method compare with other state-of-the-art methods like RAFT?***
>
> Response: We thank the reviewer for raising this question. ***Subsection 4.3 on page 8 of the revised version*** shows that the proposed KUP-BI outperforms RAFT under matched settings, using the same backbone and data splits.
>
> ***Q4. State of the art models can be improved.***
>
> Response: Thanks for your comment. To address this comment, we added a recent state-of-the-art backbone, xPatch (Stitsyuk and Choi, (2025)), to our experiments. As shown in ***Table 1 on page 7 of the revised version***, the proposed KUP-BI also brings clear improvements on top of xPatch, indicating that our method can effectively enhance strong modern forecasters.

---

> ### Author Response · Authors · 2025-11-30
> **Summary of Revisions and Responses for the Area Chair (Part 4)**
>
> ***Response to Reviewer 1s5C (Rating: 0)***
>
> First, we would like to clarify that some of Reviewer 1s5C's concerns seem to come from a misunderstanding of our setting. We do not mix prior and posterior distributions, and we do not violate any probability axioms. Our method simply adds a simple non-parametric mechanism that builds a train-only continuation library and extracts continuation-style information as auxiliary input to standard forecasting backbones. This misunderstanding has largely been resolved during the discussion phase. For example, ***Reviewer 1s5C*** noted: “I will up my score a bit, as I now see that there is something being done here that might potentially make some sense.” This indicates that the basic idea of our method is now clearer. However, Reviewer 1s5C still expresses concern about whether our approach truly operates under a fair and clearly specified data usage scheme, in particular that there is no leakage and that competing methods see the same underlying data. We do not want to give up on this direction; instead, we have made substantial revisions to the manuscript to clarify the assumptions, the train/validation/test splits, and the construction and use of the retrieval library, and we believe that we have addressed the majority of the eight detailed points raised in that review.
>
>
> ***Q1: This is presented as a nonparametric approach that distils the information from the historic library data to help with prediction, dropping all the "predicted futures" stuff as that is really confusing. This should also relate to the meta-learning literature to which this approach is related. (comment titled “Continued”, 17 Nov 2025, 03:56).***
>
> Response: We apologize for any confusion caused by our original wording. Following the reviewer's suggestion, we have removed the “predicted futures” phrasing in the revised version and now describe our approach as a non-parametric library-based method. In addition, we have renamed “Retrieved Future Prior” to “Retrieved Continuation-style Auxiliary.” Indeed, this method is related to meta-learning, and we have added a description of meta-learning in Subsection 2.2 of the revised manuscript.
>
> ***Q2: Request for a discussion of the statistical properties (consistency, bias, variance, exchangeability) of the non-parametric retrieval component.*** (Original comment: “The paper analyses the statistical properties of the non-parametric method in terms of its exchangeability properties, its consistency, bias, variance etc.”, comment titled “Continued”, 17 Nov 2025, 03:56.)
>
> Response: We thank the reviewer for this helpful and constructive guidance. A full statistical analysis of the complete KUP-BI architecture (with ratio operators, top-k retrieval, encoders and gating) would require strong additional assumptions and substantial space, which is beyond what we can reasonably include in this conference paper. In this version, we therefore do not attempt a full theoretical analysis of the main method; instead, in ***Appendix J on page 21 of the revised manuscript***, we add a simple linear continuation model where the non-parametric component is isolated and can be analyzed using standard kernel regression theory.
>
>
> ***Q3. Please justify each stage in the feature construction pipeline, and explain why you use an explicit hand-designed summary of the history library instead of a more flexible learned model (for example, a neural network).*** (Original comment: “Each stage of feature building is analysed and justified. Also why in this day is an explicit hand-chosen (and what appears fairly arbitrary) summarisation of the historic library done rather than an optimizable flexible e.g. neural model of the history?”, comment titled “Continued”, 17 Nov 2025, 03:56.)
>
> Response: Thanks for your comment.
> Regarding the request to justify each stage in the feature construction pipeline, we have added ablation studies in ***Subsection 4.4 on page 8 of the revised version***. The full results show that removing any individual stage consistently hurts performance, so each stage makes a positive contribution and the design is not arbitrary. In addition, the revised manuscript provides a brief explanation of the role of each stage in the feature construction pipeline.
>
> Regarding the choice of an explicit hand-designed summary of the history library instead of a learned neural model, our main goal in this paper is to identify and exploit a new information source, the continuation segment in training trajectories, and to open up a new perspective for the community. We therefore intentionally implement KUP-BI with a simple, hand-designed mechanism, so that the gains cannot be attributed to a large extra neural module. This simple and transparent design is also easier to plug into different backbones. Designing a more flexible learned retriever on top of this idea is an interesting direction for future work.

---

> ### Author Response · Authors · 2025-11-30
> **Summary of Revisions and Responses for the Area Chair (Part 5)**
>
> ***Response to Reviewer 1s5C (Rating: 0)***
>
> ***Q4. Is the method intended as an explanation approach, and if not, what is the actual goal of the paper and how do the constructed features relate to future information?***  (Original comment: “The paper explains (not using predicted futures, as that is the opposite of an explanation as that shouldn't help at all), why the features that are computed are the key relevant information to be summarised.”, comment titled “Continued”, 17 Nov 2025, 03:56.)
>
> Response: We thank the reviewer for this insightful question. It is important to clarify that our method is not intended as an explanation approach. Instead, the goal of the paper is to improve forecasting accuracy by augmenting standard backbones with a training-only future continuation signal. Concretely, these constructed features are computed from history–target–continuation chains in the training library and summarize how the continuation segment typically behaves relative to its history (for example, whether it tends to keep rising, flatten out, or revert). At test time, we retrieve similar histories in the library and reuse their continuation-based features as auxiliary inputs, so the constructed features carry future information learned from training data, but never from the unknown future of the test sample. In the revised version, we also clarify this point in the ***“Structure Overview” paragraph on page 4***.
>
> ***Q5. Please state all assumptions clearly, in particular any conditional stationarity assumption that similar histories in the training library and in the current forecasting scenario tend to have similar continuations.*** (Original comment: “that the assumptions are clear: there seems to be an explicit assumption of some level of conditioning stationarity between the library and the current forecasting scenario (similar histories lead to similar futures).”, comment titled “Continued”, 17 Nov 2025, 03:56.)
>
> Response: Thanks for your comment. In the revised version, we make this assumption explicit ***on page 5***. We state that we assume a mild form of conditional stationarity, namely that similar histories in the training library tend to have similar continuations in the forecasting scenario.
>
>
> ***Q6. Please ensure that all competing methods are given the same information as KUP-BI.*** (Original comment: “that the competing methods are given all the information that you use. The competing methods are allowed to train on all the data (library and history) if they are not at the moment.”, comment titled “Continued”, 17 Nov 2025, 03:56.)
>
>
> Response: Thanks for your comment. We have ensured that all competing methods are given the same information as KUP-BI.
>  All competing methods (for example, RAFT) are trained on exactly the same raw train/validation/test splits as the KUP-BI models, so they have access to the same underlying data. We clearly stated this in ***Subsection 4.3 on page 8 of the revised manuscript***.
>
>
>
> ***Q7. Please describe the training pipeline and data usage very clearly, including how the train/validation/test splits are used and how the retrieval library is built, so that there is no risk of data leakage or unfair use of information.*** (Original comment: “that the way the training pipeline is done and how the data is used is very explicit.”, comment titled “Continued”, 17 Nov 2025, 03:56.)
>
> Response: We appreciate the reviewer's comment and apologize for not clearly stating this in the original version. We only extract additional information from the training set, and all backbones and baselines are trained on exactly the same train/validation/test splits. The retrieval library is just a re-indexing of the same training trajectories (history–target–continuation) and does not include any validation or test data, so there is no leakage. At test time, each query history only accesses this train-only library to construct ratio features, which are then fused with the backbone representation. In the revised version, we make the training pipeline and data usage explicit.
>
> ***Q8. Please provide a simpler variant in the same spirit as the current method, but with a mathematically coherent formulation.*** (Original comment: “do something simpler and different in the same vein that is a mathematically coherent method, because the current approach is not.”, comment titled “Continued”, 17 Nov 2025, 03:56.)
>
> Response: We thank the reviewer for this suggestion. In the revised version, we provide a simpler variant in the same spirit that is mathematically coherent. The details of this variant are presented in ***Appendix J on page 21***.

---

### Note · Authors · 2026-01-20

I have read and agree with the venue's withdrawal policy on behalf of myself and my co-authors.